# Receptor-interacting protein kinase 2 (RIPK2) stabilizes c-Myc and is a therapeutic target in prostate cancer metastasis

Yiwu Yan[1,11], Bo Zhou[1,9,11], Chen Qian[1], Alex Vasquez[1], Mohini Kamra[2], Avradip Chatterjee[3], Yeon-Joo Lee [1], Xiaopu Yuan[4], Leigh Ellis [3,5,6], Dolores Di Vizio[1,3,4], Edwin M. Posadas[5,6], Natasha Kyprianou [7], Beatrice S. Knudsen [3,4,6,10], Kavita Shah [2], Ramachandran Murali [3,6], Arkadiusz Gertych [1,4], Sungyong You [1,3,6], Michael R. Freeman [1,3,6,8] & Wei Yang [1,3,6,8✉]

Despite progress in prostate cancer (PC) therapeutics, distant metastasis remains a major cause of morbidity and mortality from PC. Thus, there is growing recognition that preventing or delaying PC metastasis holds great potential for substantially improving patient outcomes. Here we show receptor-interacting protein kinase 2 (RIPK2) is a clinically actionable target for inhibiting PC metastasis. *RIPK2* is amplified/gained in ~65% of lethal metastatic castration-resistant PC. Its overexpression is associated with disease progression and poor prognosis, and its genetic knockout substantially reduces PC metastasis. Multi-level proteomics analyses reveal that RIPK2 strongly regulates the stability and activity of c-Myc (a driver of metastasis), largely via binding to and activating mitogen-activated protein kinase kinase 7 (MKK7), which we identify as a direct c-Myc-S62 kinase. RIPK2 inhibition by preclinical and clinical drugs inactivates the noncanonical RIPK2/MKK7/c-Myc pathway and effectively impairs PC metastatic outgrowth. These results support targeting RIPK2 signaling to extend metastasis-free and overall survival.

[1] Department of Surgery, Cedars-Sinai Medical Center, Los Angeles, CA, USA. [2] Department of Chemistry and Purdue University Center for Cancer Research, Purdue University, West Lafayette, IN, USA. [3] Department of Biomedical Sciences, Cedars-Sinai Medical Center, Los Angeles, CA, USA. [4] Department of Pathology and Laboratory Medicine, Cedars-Sinai Medical Center, Los Angeles, CA, USA. [5] Department of Medicine, Cedars-Sinai Medical Center, Los Angeles, CA, USA. [6] Samuel Oschin Comprehensive Cancer Institute, Cedars-Sinai Medical Center, Los Angeles, CA, USA. [7] Department of Urology, Icahn School of Medicine at Mount Sinai, New York, New York, NY, USA. [8] Department of Medicine, University of California Los Angeles, Los Angeles, CA, USA. [9] Present address: InterVenn Biosciences, South San Francisco, CA, USA. [10] Present address: Department of Pathology, University of Utah, Salt Lake City, UT, USA. [11] These authors contributed equally: Yiwu Yan, Bo Zhou. ✉email: wei.yang@cshs.org

Prostate cancer (PC) is the second most common non-skin cancer in men worldwide and causes about 360,000 deaths globally each year[1]. Despite advancements in PC treatment, distant metastasis continues to be a prominent cause of PC morbidity and mortality, inflicting considerable social and economic burdens[2,3]. Since 2018, the U.S. Food and Drug Administration (FDA) approved three androgen receptor (AR) inhibitors—apalutamide, enzalutamide, and darolutamide—for the treatment of non-metastatic castration-resistant PC (nmCRPC, also known as M0 CRPC). The approvals were based on phase III clinical trials using metastasis-free survival (MFS) as the primary endpoint for the first time[4–6]. Longer follow-up studies showed that the drugs extended the median overall survival by 10–14 months[7,8], unequivocally demonstrating that delaying PC metastasis provides strong clinical benefits. Nevertheless, to further improve MFS—a strong surrogate of overall survival[9]—novel therapeutic targets and their mechanisms of action in PC metastasis need to be identified. Candidate targets with the following characteristics are particularly attractive: (1) frequent overexpression in advanced PC, (2) significant association of high expression levels with poor prognosis, and (3) availability of potent and tolerable small-molecule inhibitors. Such targets have a high potential of rapid translation into the clinic to benefit at least a significant subset of PC patients.

In the present study, through bioinformatic and functional analyses, we discover receptor-interacting protein kinase 2 (RIPK2) as one such particularly attractive target and show that RIPK2 is required for PC metastasis. RIPK2 is a Ser/Thr/Tyr kinase that has been well characterized in inflammation and innate immunity, functioning mainly via the NOD/RIPK2/NF-κB signaling pathway[10,11]. In contrast, little is known about the molecular functions of RIPK2 in the metastasis of PC and other cancer types. Distinct from the canonical NOD/RIPK2/NF-κB signaling pathway, we demonstrate that RIPK2 potently stabilizes and activates c-Myc (encoded by *MYC*), largely by binding to and activating MKK7, which we discover is a direct c-Myc-S62 kinase. Notably, many studies have shown that *MYC* overexpression drives the progression and metastasis of many cancer types including PC[12–17], and that *MYC* knockdown substantially inhibits the metastasis of PC and other cancers[18–20]. Nevertheless, no direct c-Myc-targeting drugs have ever been approved by the FDA, largely because of the non-enzymatic nature of c-Myc and potential toxicity associated with its inhibition, due to expression in normal tissues[21,22]. Conversely, RIPK2/MKK7/c-Myc pathway inhibition by GSK583 (a preclinical RIPK2-selective inhibitor) or ponatinib (an existing FDA-approved drug) suppressed PC cell invasion and colony formation, inhibited PC metastatic outgrowth, and did not cause discernable toxicity in mice. Thus, RIPK2 is credentialed as a promising therapeutic target in PC metastasis.

## Results

### RIPK2 is a top candidate druggable target for PC metastasis.
To identify therapeutically viable protein targets for metastasis in a substantial subset of PC patients, we applied stringent criteria to filter three large-scale clinical omics databases: the cBioPortal for Cancer Genomics[23], the Prostate Cancer Transcriptome Atlas (PCTA)[24], and the Pharos for human druggable genome[25] (see the "Public Resources" section in Methods). These criteria include: (1) genes are recurrently (>10%) amplified in lethal metastatic castration-resistant PC (mCRPC)[26–28], (2) mRNA levels have a high correlation (rho > 0.9 and $p < 0.01$) with Gleason score and the natural history of PC progression, which are associated with increased metastatic risk, and (3) proteins can be readily targeted by $T_{clin}$- or $T_{chem}$-grade inhibitors, which are

FDA-approved or have an activity cutoff of <30 nM[25]. A total of 574, 1643, and 2208 human genes meet the three criteria, respectively (Fig. 1a). Seven genes meet all three criteria, representing candidate druggable targets of PC metastasis (Fig. 1a). Except for PMVK, these genes are amplified in 10–26% of PC tissue specimens in at least two out of the three mCRPC cohorts (Supplementary Fig. 1a). Notably, among the seven genes, *RIPK2* has the highest mRNA level increase with progression from benign to lethal disease (Fig. 1b).

To determine whether RIPK2 is associated with advanced PC at different molecular levels (i.e., DNA, RNA, and protein), we analyzed publicly accessible PC genomics and transcriptomics data sets and performed immunohistochemical (IHC) analysis of PC tissue microarrays. At the DNA level, the frequency of *RIPK2* copy number amplification or gain increases along with PC progression, ranging from 12.3% in low-grade, low-risk PC to 64.6% in mCRPC (Fig. 1c and Supplementary Fig. 1b). *RIPK2* copy numbers are strongly associated with *RIPK2* mRNA expression levels in PC tissue specimens ($r = 0.68$, $p = 2.2E–16$) (Supplementary Fig. 1c). As expected, the frequency of *RIPK2* mRNA overexpression also increases with PC progression, rising from 0.8% in tumor-adjacent normal tissue to 45.4% in metastatic or castration-resistant PC (Fig. 1d and Supplementary Fig. 1d). Furthermore, *RIPK2* mRNA abundance is strongly associated with RIPK2 protein abundance in various human cancer cell lines ($r = 0.66$, $p = 2.2E–16$) (Supplementary Fig. 1e). Consistently, the frequency of RIPK2 protein overexpression is positively associated with Gleason scores (a predictor of PC recurrence), ranging from 3.7% in tumor-adjacent normal tissue to 26.7% in tumors with the Gleason score of 9 (Fig. 1e and Supplementary Fig. 1f, g). Consistent with the clinical findings, RIPK2 protein abundance is substantially higher in PC cell line models capable of forming metastases in immunocompromised mice, such as PC3, DU145, 22Rv1, and LNCaP, compared with non-metastatic prostate cell lines such as RWPE-2 and RWPE-1 (Fig. 1f). Collectively, the findings suggest that the genetic alteration and overexpression of RIPK2 are strongly associated with PC progression and metastasis and are frequent events in lethal or high-risk PCs.

To investigate whether RIPK2 is associated with PC aggressiveness, we compared *RIPK2* mRNA levels in PC subtypes with different levels of aggressiveness. *RIPK2* mRNA levels are significantly higher in aggressive PC subtypes such as Prostate Cancer Subtype 1 (PCS1) and Luminal B (LumB)[24,29], compared with the other two less aggressive PCS or PAM50 (prediction analysis of microarray 50) subtypes, in both PCTA and The Cancer Genome Atlas (TCGA) Firehose Legacy cohorts (Fig. 1g and Supplementary Fig. 1h, respectively). PCS1 tumors progress more rapidly to metastatic disease than PCS2 and PCS3 tumors (hazard ratio = 4.8)[24], and LumB tumors exhibit poorer clinical prognosis than LumA and Basal tumors[29].

To determine whether RIPK2 is associated with poor prognosis, we performed Kaplan–Meier survival analyses. The high mRNA abundance of *RIPK2* is significantly associated with worse progression-free and metastasis-free survival of PC patients (Fig. 1h, i, respectively). In addition to PC, *RIPK2* is frequently amplified and/or overexpressed in several other cancer types (Supplementary Fig. 2), and its mRNA overexpression is associated with significantly shorter overall survival in nine cancer types (Supplementary Fig. 3). Thus, *RIPK2* overexpression is associated with poor prognosis in PC and many other cancers.

### RIPK2 is required for PC metastasis.
To investigate whether targeting RIPK2 suppresses CRPC progression and metastasis, we stably knocked out *RIPK2* from three mCRPC cell lines using the CRISPR/Cas9 system[30,31] (Fig. 2a). Of note, PC3 has

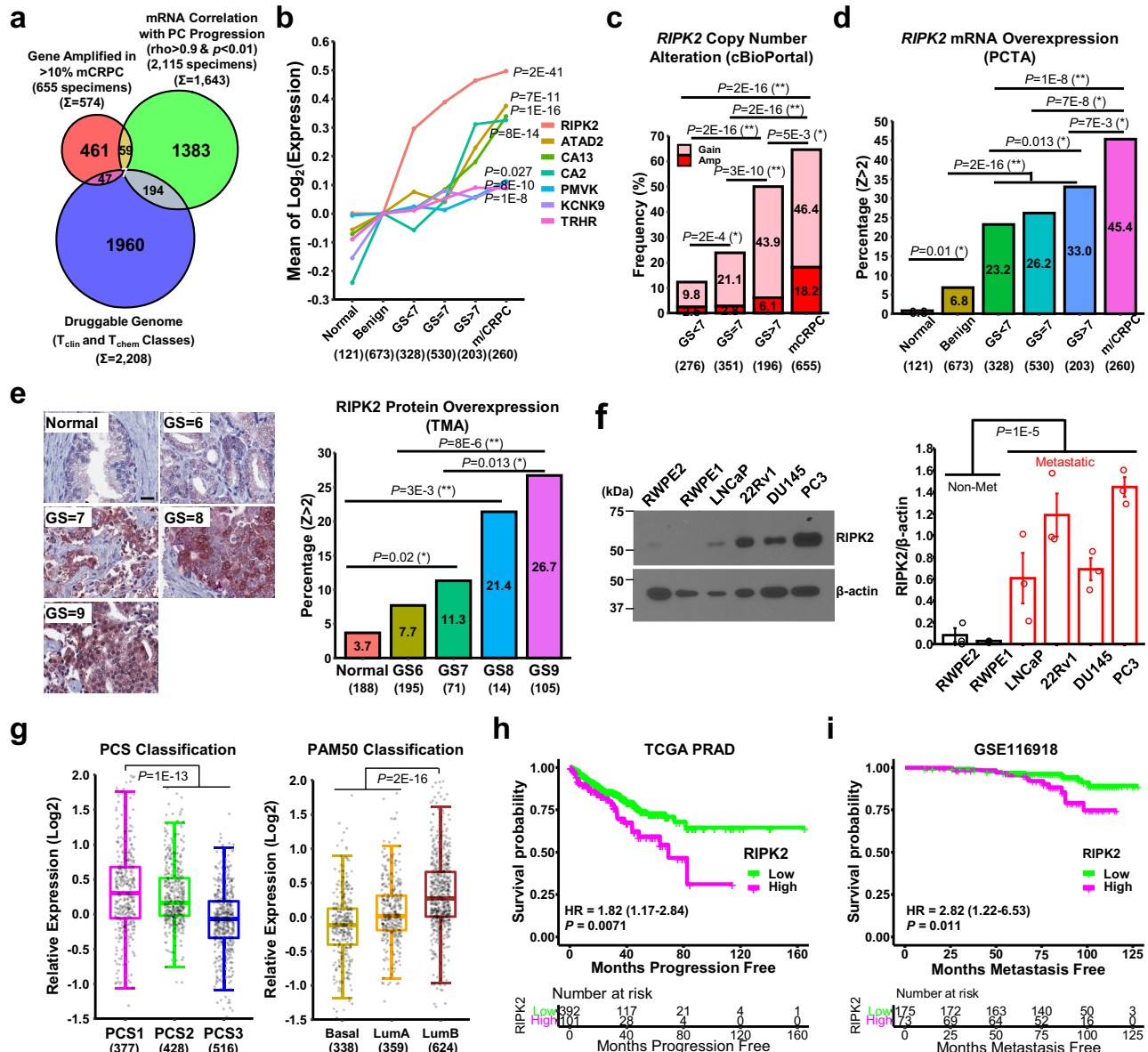

**Fig. 1 RIPK2 is a top candidate druggable target for prostate cancer (PC) metastasis. a** Venn diagram of human genes meeting the three indicated criteria. **b** mRNA level changes of the seven overlapping genes along with PC progression. GS: Gleason score. **c** Bar plot of frequencies of *RIPK2* copy number amplification and gain along with PC progression. **d** Bar plot of frequencies of *RIPK2* mRNA overexpression ($Z > 2$ relative to tumor-adjacent normal tissue) along with PC progression. PCTA: Prostate Cancer Transcriptome Atlas. **e** Comparison of RIPK2 protein expression levels in primary PC. Left panel: representative IHC images (scale bar = 25 μm). Right panel: frequencies of RIPK2 protein overexpression ($Z > 2$ relative to tumor-adjacent normal tissue) along with PC progression. TMA: Tissue microarray. **f** Comparison of RIPK2 protein levels in six commonly used prostate cell line models. Left panel: representative immunoblotting images. Right panel: bar plot of relative RIPK2 protein abundance normalized against β-actin ($n = 3$ biologically independent samples per group). Non-met: non-metastatic. **g** Box plots of *RIPK2* mRNA levels in three PCS (left) and PAM50 (right) subtypes of prostate tumors in the PCTA cohort. Box plots indicate median (middle line), 25th and 75th percentile (box), and 5th and 95th percentile (whiskers). PCS: prostate cancer subtype; PAM50: prediction analysis of microarray 50. **h** Kaplan–Meier progression-free survival analysis of PC patients in the TCGA PanCancer Atlas cohort stratified by *RIPK2* mRNA abundance. TCGA: The Cancer Genome Atlas; PRAD: prostate adenocarcinoma; HR: hazard ratio (95% confidence interval). **i** Kaplan–Meier metastasis-free survival analysis of PC patients in the Jain cohort (GSE116918) stratified by *RIPK2* mRNA abundance. The numbers in parentheses represent sample sizes. Nominal p-values were determined by one-way ANOVA test (**b**), one-sided chi-square test (**c-e**), unpaired two-tailed Student's t-test (**f**), two-sided rank-sum test (**g**), or two-sided log-rank test (**h**, **i**). Effect sizes were calculated using Cramer's V and presented as * weak (<0.2) or ** moderate (<0.6) (**c-e**). Data are mean ± standard error of the mean (SEM) (**f**).

*RIPK2* gene amplification (5-6 copies/cell), and DU145 and 22Rv1 have *RIPK2* gain (~3 copies/cell)[32,33]. All three cell lines have much higher RIPK2 protein levels than non-metastatic prostate cell lines (Fig. 1f). *RIPK2*-KO only had a negligible effect on cell proliferation or migration in 2D culture (Supplementary Fig. 4a, b). Nevertheless, *RIPK2*-KO by guide RNA (gRNA)

#10 significantly reduced the Matrigel invasion of PC cells (Fig. 2b). In addition, *RIPK2*-KO by a different gRNA also achieved significant reductions of PC cell invasion (Supplementary Fig. 4c), excluding possible off-target effects of CRISPR/Cas9 knockout. Moreover, *RIPK2*-KO significantly suppressed 3D spheroid invasion (Fig. 2c and Supplementary Fig. 4d),

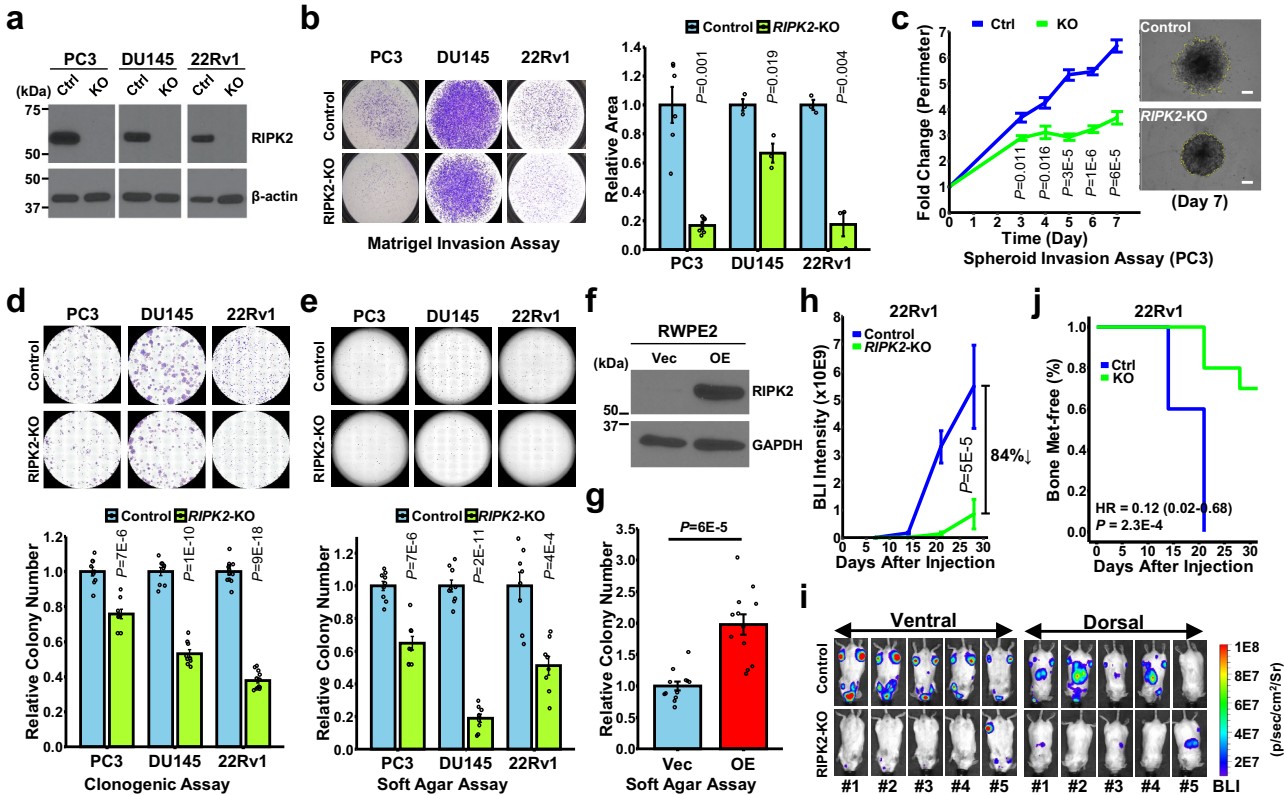

**Fig. 2 RIPK2 is required for PC metastasis. a** Representative immunoblots of RIPK2 in total lysates derived from control PC cells and cells with *RIPK2* knockout by CRISPR/Cas9 using gRNA10. Experiments were repeated twice independently with similar results. Ctrl: control; KO: knockout. **b** Representative images (left) and quantification (right) of Matrigel invasion assay of control or *RIPK2*-KO PC cells. $N = 6, 3, 3$ biologically independent samples per group for PC3, DU145, and 22Rv1 cells, respectively. **c** Quantification and representative images of 3D spheroid invasion assay of control or *RIPK2*-KO PC3 cells ($n = 6$ biologically independent samples per group). Scale bars, 300 μm. **d, e** Representative images (upper) and quantification (lower) of **d** anchorage-dependent or **e** anchorage-independent colony formation assay of control or *RIPK2*-KO PC cells. $N = 9, 9, 12$ biologically independent samples per group for PC3, DU145, and 22Rv1 cells, respectively (**d**). $N = 9, 9, 8$ biologically independent samples per group for PC3, DU145, and 22Rv1 cells, respectively (**e**). **f** Representative immunoblots of the indicated proteins in total lysates derived from control and *RIPK2*-overexpressing (OE) RWPE-2 cells. Experiments were repeated twice independently with similar results. Vec: vector control; OE: overexpression. **g** Quantification of anchorage-independent colony formation assay of control and *RIPK2*-OE RWPE-2 cells ($n = 12$ biologically independent samples per group). **h–j** Luciferase-tagged control or *RIPK2*-KO 22Rv1 cells were intracardially injected into male SCID/Beige mice, which were monitored every week for 4 weeks by in vivo bioluminescence imaging (BLI). **h** Total BLI intensities over time ($n = 5$ biologically independent mice per group). **i** BLI images taken 4 weeks after the i.c. injection from the ventral and dorsal sides. **j** Percentages of knee joints free of metastasis ($n = 10$ biologically independent joints per group) over time. Nominal *p*-values were determined by unpaired two-tailed Student's *t*-test (**b–e, g**), two-way ANOVA (**h**), or two-sided log-rank test (**j**). Data are shown as mean ± SEM (**b–e, g, h**).

anchorage-dependent colony formation (Fig. 2d), and anchorage-independent soft agar colony formation (Fig. 2e). In addition, ectopic expression of RIPK2 in RWPE-2 cells significantly increased their anchorage-independent colony formation (Fig. 2f, g). Together, these results indicate that RIPK2 is required for PC cell invasion and colony formation, key biological processes in cancer metastasis.

To investigate whether *RIPK2*-KO suppresses PC metastasis in vivo, we conducted intracardiac (*i.c.*) injection of luciferase-labeled control and *RIPK2*-KO 22Rv1 cells into male SCID/Beige mice (Supplementary Fig. 5). 22Rv1 cells express both full-length and constitutively active variants of AR, recapitulating aggressive mCRPC tumors[34]. Compared with the control group, mice harboring *RIPK2*-KO 22Rv1 cells showed substantially lower metastatic burden (84% reduction at week 4) (Fig. 2h, i and Supplementary Fig. 6a, b), longer bone metastasis-free survival (Fig. 2j), and less weight loss (Supplementary Fig. 6c), suggesting that RIPK2 is critical for PC metastasis. In addition, to determine whether *RIPK2*-KO affects tumor growth in vivo, we performed a subcutaneous injection of control and *RIPK2*-KO 22Rv1 cells in

male SCID/Beige mice. Consistent with the in vitro cell proliferation results (Supplementary Fig. 4a), *RIPK2*-KO 22Rv1 xenografts had a significant but modest reduction in tumor growth rate, weight, and size, compared with control tumors (Supplementary Fig. 7). Collectively, the results suggest that RIPK2 is mainly required for PC metastasis rather than tumor growth.

**Label-free proteomics reveals RIPK2 as a potent activator of c-Myc.** In the canonical RIPK2 signaling pathway, RIPK2 mediates the signaling from NOD to NF-κB, resulting in the transcription of inflammatory cytokines[10,11]. However, *RIPK2*-KO did not inhibit NF-κB signaling in PC3 or 22Rv1 cells (Supplementary Fig. 8), suggesting that RIPK2 may function via a non-canonical signaling pathway in PC. To map the non-canonical RIPK2 pathway, we performed an unbiased label-free proteomic analysis to comprehensively identify RIPK2 downstream effectors, followed by bioinformatic analysis to identify key mediators of RIPK2 signaling (Supplementary Fig. 9). Here, we chose the

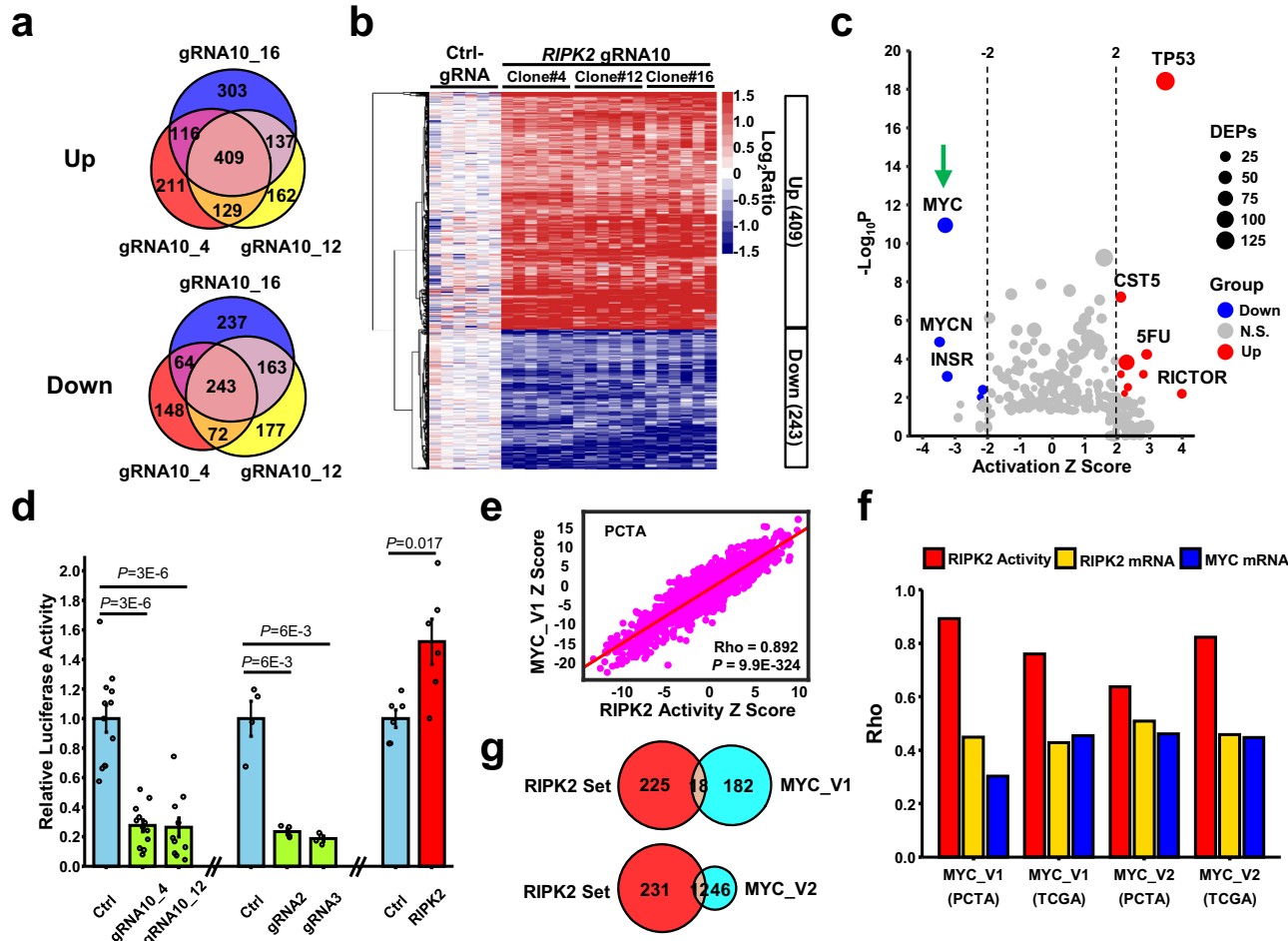

**Fig. 3 Label-free proteomics reveals RIPK2 as a potent activator of c-Myc. a** Venn diagrams of protein groups that are upregulated (upper) or downregulated (lower) in the three *RIPK2*-KO PC3 single-cell clones (#4, #12, and #16), compared with control cells. **b** Heatmap of the 652 overlapping differentially expressed protein groups in the control or *RIPK2*-KO PC3 clones (3 biological replicates × 2 LC-MS/MS runs). **c** Scatter plot of putative regulators upstream of the 652 differentially expressed proteins. The activation *Z* scores and p-values were computed by Ingenuity Pathway Analysis. **d** Bar plot of the relative *MYC* luciferase reporter activity in PC3 cells under the indicated conditions ($n = 12$, 4, and 6 biologically independent samples in the left three, middle three, and right two conditions, respectively). Data are shown as mean ± SEM. **e** Scatter plot of RIPK2-induced activity *Z* scores against Hallmark_MYC_Targets_V1 gene-set activity *Z* scores in the PCTA cohort. **f** Bar plot of the Spearman's correlation coefficients (rho) of *RIPK2*-induced activity scores (red), *RIPK2* mRNA levels (orange), and *MYC* mRNA levels (blue) with *MYC* activity scores (V1 or V2) in the PCTA or TCGA (Firehose Legacy) PC cohorts. **g** Venn diagram of genes contained in the RIPK2-induced protein signature (i.e., the 243 protein groups downregulated by *RIPK2*-KO) and those in the Hallmark_MYC_Targets_V1 (upper) or _V2 (lower) gene set. Nominal p-values were determined by two-sided Fisher's exact test (**c**), unpaired two-tailed Student's *t*-test (**d**), or using a *t*-distribution with $n$-2 degrees of freedom (**e**).

control and *RIPK2*-KO PC3 cells for comparison, because PC3 has the highest RIPK2 protein expression level among all the cell lines screened (Fig. 1f). To identify consistent changes across different clones, three PC3 single-cell clones (#4, #12, and #16) with stable *RIPK2*-KO were isolated and expanded, and their proteomes were compared with those of control PC3 cells.

Proteomic analysis identified 5237 protein groups with a false discovery rate (FDR) of <1%. In bottom-up proteomics analysis, different proteins identified by the same set of shared peptides cannot be distinguished, so they are collapsed into a "protein group" to minimize redundant identifications. After quality assessment (Supplementary Fig. 10) and statistical analysis (Supplementary Fig. 11), we found that 243 and 409 protein groups are consistently downregulated or upregulated in all the three *RIPK2*-KO PC3 clones, compared with control PC3 cells, respectively ($q < 0.005$) (Figs. 3a, b and Supplementary Data 1 and 2). ToppFun gene ontology enrichment analysis showed that the downregulated protein set is the most significantly associated with ribosome biogenesis and mitochondria, whereas the

upregulated protein set is the most significantly associated with cytoskeleton organization and focal adhesion (Supplementary Fig. 12). Using the downregulated and upregulated protein sets, we computed the *Z* scores of RIPK2-induced and RIPK2-repressed activities, respectively, in each sample of the PCTA and TCGA cohorts. In line with the aforementioned *RIPK2* mRNA findings (Fig. 1b, g), the RIPK2-induced activity scores are positively—whereas the RIPK2-repressed activity scores are negatively—associated with PC progression and aggressiveness in both cohorts (Supplementary Fig. 13).

To identify key mediators of RIPK2 signaling in PC, we conducted an upstream regulator analysis of the 652 (243 downregulated and 409 upregulated) differentially expressed protein groups (DEPs), using Ingenuity Pathway Analysis (IPA). Among all putative upstream regulators, c-Myc was the most significantly suppressed by *RIPK2*-KO (Fig. 3c). Consistently, gene-set enrichment analysis (GSEA) of all quantified proteins identified MYC_Targets_V1 and V2 as the most strongly downregulated hallmark gene signatures in *RIPK2*-KO PC3

clones, compared with control cells (Supplementary Fig. 14). Thus, c-Myc is potentially a central mediator of RIPK2 signaling. To confirm that RIPK2 activates c-Myc, we performed c-Myc luciferase reporter assays in *RIPK2*-manipulated PC3 cells. *RIPK2*-KO by three different gRNAs substantially decreased—whereas ectopic expression of human *RIPK2* significantly increased—c-Myc activity in PC3 cells, compared with their respective controls (Fig. 3d).

To evaluate the clinical relevance of our finding that RIPK2 activates c-Myc, we performed correlation analyses in PC tissue specimens. To compute activity $Z$ scores, we used the genes encoding the 243 protein groups downregulated by *RIPK2*-KO as RIPK2 signature genes. We also retrieved the Hallmark_MYC_-Targets_V1 and _V2 gene sets from the Molecular Signatures Database (MSigDB)[35] and used them as MYC_V1 and MYC_V2 signature genes, respectively. Strikingly, in both PCTA and TCGA cohorts, *RIPK2*-induced activity scores are highly correlated with *MYC* activity scores (Fig. 3e, f and Supplementary Fig. 15a), even though the overlap between the *RIPK2* and *MYC* signature genes is modest (Fig. 3g). In comparison, *RIPK2* mRNA levels have lower correlations with *MYC* activity scores, but the correlation coefficients are still comparable to those between *MYC* mRNA levels and *MYC* activity scores (Fig. 3f and Supplementary Fig. 15a). Interestingly, among all the human kinases, *RIPK2* mRNA levels have the strongest correlation with *MYC* activity scores in the PCTA and TCGA cohorts on average (Supplementary Fig. 15b). Out of the 50 Hallmark gene sets in the MSigDB[35], the *MYC*_Targets gene sets (V1 and V2) are among the most strongly associated with RIPK2 mRNA and activity levels in both cohorts (Supplementary Fig. 16). In addition, our pan-cancer analysis showed that *RIPK2*-induced activity scores are strongly (rho = 0.88 on average) correlated with *MYC* activity scores across all the 32 cancer types (Supplementary Fig. 17). Taken together, the findings suggest that RIPK2 is a potent activator of c-Myc in PC and potentially in other cancers.

**RIPK2 upregulates the c-Myc protein by phosphorylating c-Myc-S62 and preventing c-Myc from proteasomal degradation**. Studies have shown that the c-Myc oncoprotein is necessary or sufficient for cancer cell invasion[36–38], anchorage-dependent colony formation[38–40], anchorage-independent colony formation[41–45], and metastasis[12–20]. Thus, we investigated whether *RIPK2*-KO suppresses PC metastasis via modulating the c-Myc protein. Our results showed that stable *RIPK2*-KO by two independent gRNAs substantially downregulated c-Myc protein levels in PC3, DU145, and 22Rv1 cells (Fig. 4a). Ectopic expression of *RIPK2m4*, whose gRNA10-targeting spacer region was silently mutated to disable Cas9 recognition, increased the protein abundance of endogenous c-Myc in a dose-dependent fashion in *RIPK2*-KO PC3 cells (Fig. 4b) and of exogenous c-Myc in five different cell lines (Fig. 4c). Taken together, the findings suggest that RIPK2 positively regulates the c-Myc protein independent of cell type.

The *RIPK2* and *MYC* genes are located about 38 mega-bases apart on chromosome 8q (Fig. 4d), whose gains are among the most frequent cytogenetic alterations in PC[46]. Our analysis of three large-scale genomic profiling studies of mCRPC tissue specimens ($n = 655$) showed that *RIPK2* and *MYC* are co-amplified/gained in 60.6% of all specimens (Fig. 4e). Frequent *RIPK2* and *MYC* co-amplification/gain also occurs in several other cancer types (Supplementary Fig. 18a). Interestingly, compared with the forced expression of *RIPK2* or *MYC* alone, the co-transfection of both *RIPK2* and *MYC*—a mimic of *RIPK2* and *MYC* co-amplification/gain—resulted in much higher c-Myc protein levels in *RIPK2*-KO DU145, 22Rv1, and HEK293T cells

(Fig. 4f and Supplementary Fig. 18b). Thus, the findings indicate that the co-amplification/gain of *RIPK2* and *MYC*, a frequent event in PC and several other cancer types, synergistically contributes to c-Myc protein abundance independent of the cellular context.

Using quantitative real-time polymerase chain reaction (RT-PCR), we found that transient *RIPK2* overexpression in PC3 cells, *RIPK2*-KO in 22Rv1 cells, or RIPK2 inhibition by GSK583 (a RIPK2-selective inhibitor) at different doses and time only modestly (~10%) regulated *MYC* mRNA abundance (Supplementary Fig. 19). In comparison, *RIPK2*-KO in PC3 cells caused a stronger reduction in *MYC* mRNA levels (by ~47%) (Supplementary Fig. 19a). However, this is likely a secondary event, considering that transient *RIPK2*-OE or RIPK2 inhibition in PC3 cells only marginally modulated *MYC* mRNA levels (Supplementary Fig. 19). Collectively, the regulation of c-Myc protein abundance by RIPK2 is mainly via a post-transcriptional mechanism.

In human cells, the c-Myc protein can be rapidly stabilized through S62 phosphorylation, which prevents the targeted degradation of c-Myc by the ubiquitin-proteasome system[47]. Ectopic overexpression of RIPK2 increased the abundance of phospho-c-Myc-S62 (p-c-Myc-S62) and total c-Myc in a dose-dependent fashion (Fig. 4g). Moreover, the abundance of p-c-Myc-S62 increased much faster than the mRNA and protein abundance of c-Myc (Fig. 4h), suggesting that RIPK2 upregulates c-Myc largely (if not exclusively) by phosphorylating and stabilizing c-Myc. Consistently, *RIPK2*-KO significantly increased the K48-linked ubiquitination of c-Myc, a signal for proteasomal degradation[47] (Fig. 4i). In addition, *RIPK2*-KO reduced the half-life of the c-Myc protein, whose degradation can be blocked by the proteasome inhibitor MG132 (Fig. 4j and Supplementary Fig. 20). RIPK2 is more frequently amplified or gained in mCRPC or overexpressed in primary PC than all known direct c-Myc-S62 kinases (Supplementary Fig. 21). This raises a possibility that RIPK2 can more efficiently stabilize and activate c-Myc, and thus its overexpression is more favored by PC cells than the known direct c-Myc-S62 kinases. Collectively, the results suggest that RIPK2 stabilizes c-Myc by either directly or indirectly phosphorylating its S62 residue and preventing it from proteasomal degradation, and that RIPK2 is potentially a major stabilizer of c-Myc in PC cells.

**Integrative proteomics analysis identifies candidate kinase pathways mediating RIPK2 phosphorylation of c-Myc-S62**. To determine whether RIPK2 directly phosphorylates c-Myc under physiological conditions, we investigated whether the two proteins bind to each other—a prerequisite for direct RIPK2 phosphorylation of c-Myc—by performing proximity ligation assay (PLA) and immunoprecipitation (IP). Both failed to detect the protein-protein association between endogenous RIPK2 and c-Myc (Supplementary Fig. 22), suggesting that RIPK2 phosphorylates c-Myc-S62 via intermediary kinase(s). To identify these, we first mapped the subcellular location and functional domain(s) of RIPK2 that are critical for the regulation of c-Myc. The RIPK2 protein is predominantly localized in the cytoplasm but was also detected in nuclei under certain conditions[48,49]. RIPK2 contains two functional domains: an N-terminal kinase domain and a C-terminal CARD domain[10]. To determine which subcellular localization and functional domain(s) are critical for regulation of c-Myc, we generated five C-terminally 3×FLAG-tagged RIPK2 constructs: (1) wild-type, (2) N-terminally tagged with a nuclear localization signal (NLS), (3) N-terminally tagged with a nuclear export signal (NES), (4) with the deletion of the kinase domain (ΔKinase), and (5) with the deletion of the CARD

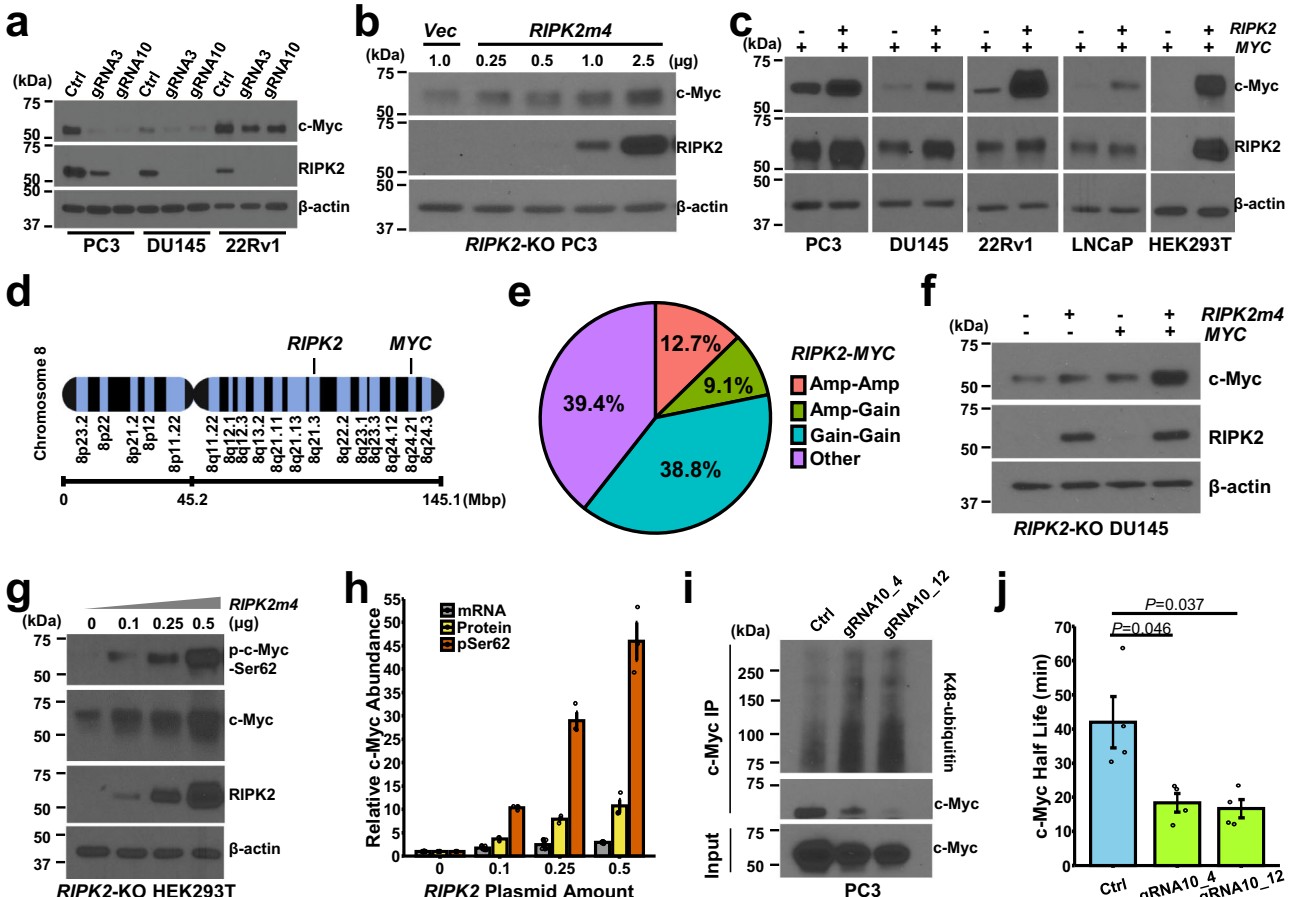

**Fig. 4 RIPK2 upregulates the c-Myc protein by phosphorylating c-Myc-S62 and preventing c-Myc from proteasomal degradation. a–c** Representative immunoblots of the indicated proteins of total lysates from **a** control or *RIPK2*-KO (by gRNA3 or gRNA10) PC cells, **b** *RIPK2*-KO PC3 cells transiently transfected with the indicated amounts of the *RIPK2m4* plasmid, or **c** parental cells transiently co-transfected with vector or *RIPK2* (2 µg plasmid per cell line except for 0.5 µg for HEK293T) and *MYC* (1 µg per cell line except for 0.5 µg for HEK293T) plasmids. Experiments were repeated twice independently with similar results (**a–c**). **d** Schematic of *RIPK2* and *MYC* locations on human chromosome 8. **e** Pie chart of the percentages of mCRPC tissue specimens (*n* = 655) with *RIPK2* and *MYC* gene co-amplification/gain. **f, g** Representative immunoblots of the indicated proteins of total lysates from **f** *RIPK2*-KO DU145 cells transiently transfected with *RIPK2* (0.5 µg) and/or *MYC* (0.5 µg) plasmids or **g** *RIPK2*-KO HEK293T cells transiently co-transfected with the indicated amounts of *RIPK2* plasmid and 0.5 µg *MYC* plasmid. Experiments were repeated three times independently with similar results (**f, g**). **h** Bar plot of the relative changes of *MYC* mRNA levels, c-Myc protein levels, or p-c-Myc-S62 protein levels (*n* = 9, 3, and 3 biologically independent samples per group, respectively) in response to the ectopic expression of the indicated amounts of the *RIPK2m4* plasmid in *RIPK2*-KO HEK293T cells. **i** c-Myc was immunoprecipitated from control or *RIPK2*-KO cells (clones #4 and #12). Immunoprecipitation product and total lysates were analyzed by IB for the indicated proteins. Experiments were repeated twice independently with similar results. **j** Bar graph of the half-lives of the c-Myc protein in control or *RIPK2*-KO PC3 cells (*n* = 4 biologically independent samples per group). Nominal *p*-values were determined by unpaired two-tailed Student's *t*-test (**j**). Data are shown as mean ± SEM (**h, j**).

domain (∆CARD). Our immunofluorescence results confirmed that the NLS and NES targeted RIPK2 into the nuclei and the cytoplasm, respectively (Supplementary Fig. 23). Ectopic expression of NES-RIPK2 and not NLS-RIPK2, as well as of ∆CARD-RIPK2 but not ∆Kinase-RIPK2, significantly increased c-Myc protein abundance (Fig. 5a). These findings suggest that both the cytoplasmic localization and the kinase domain of RIPK2 are critical for the regulation of c-Myc.

Thus, we postulated that certain protein(s) binding to the kinase domain, but not other regions, of cytoplasmic RIPK2 are critical for mediating RIPK2's indirect phosphorylation of c-Myc-S62. To identify such protein(s), we performed a rigorously controlled interactome analysis by immunoprecipitation-mass spectrometry (IP-MS), including two experimental (G5 and G3) and three control (G1, G2, and G4) conditions (Fig. 5b and Supplementary Fig. 24). The analysis identified 1189 proteins with an FDR of <1%. After quality assessment (Supplementary

Fig. 25) and statistical comparison (Supplementary Fig. 26), we identified 219 protein candidates that associate with the kinase domain and no other regions of cytoplasmic RIPK2 (Fig. 5c and Supplementary Fig. 27a). These candidates include two known RIPK2-binding partners, XIAP and RPL38[50,51] (Supplementary Fig. 27b). The candidate interacting proteins also include six kinases (Supplementary Data 3), yet none was previously reported to be direct c-Myc-S62 kinases.

To identify kinase(s) that link RIPK2 to c-Myc-S62, we then performed a phosphoproteomic comparison of two *RIPK2*-KO PC3 clones with control PC3 cells (Fig. 5d). The analysis identified 6749 phosphosites, which correspond to 2716 phosphoproteins, with an FDR of <1% and a localization probability of >0.75 (Supplementary Fig. 28a). Following quality assessment and statistical analysis (Supplementary Fig. 28b, c), we analyzed relative kinase activities using the kinase-substrate enrichment analysis (KSEA)[52]. The relative activities of 50 and 58 kinases

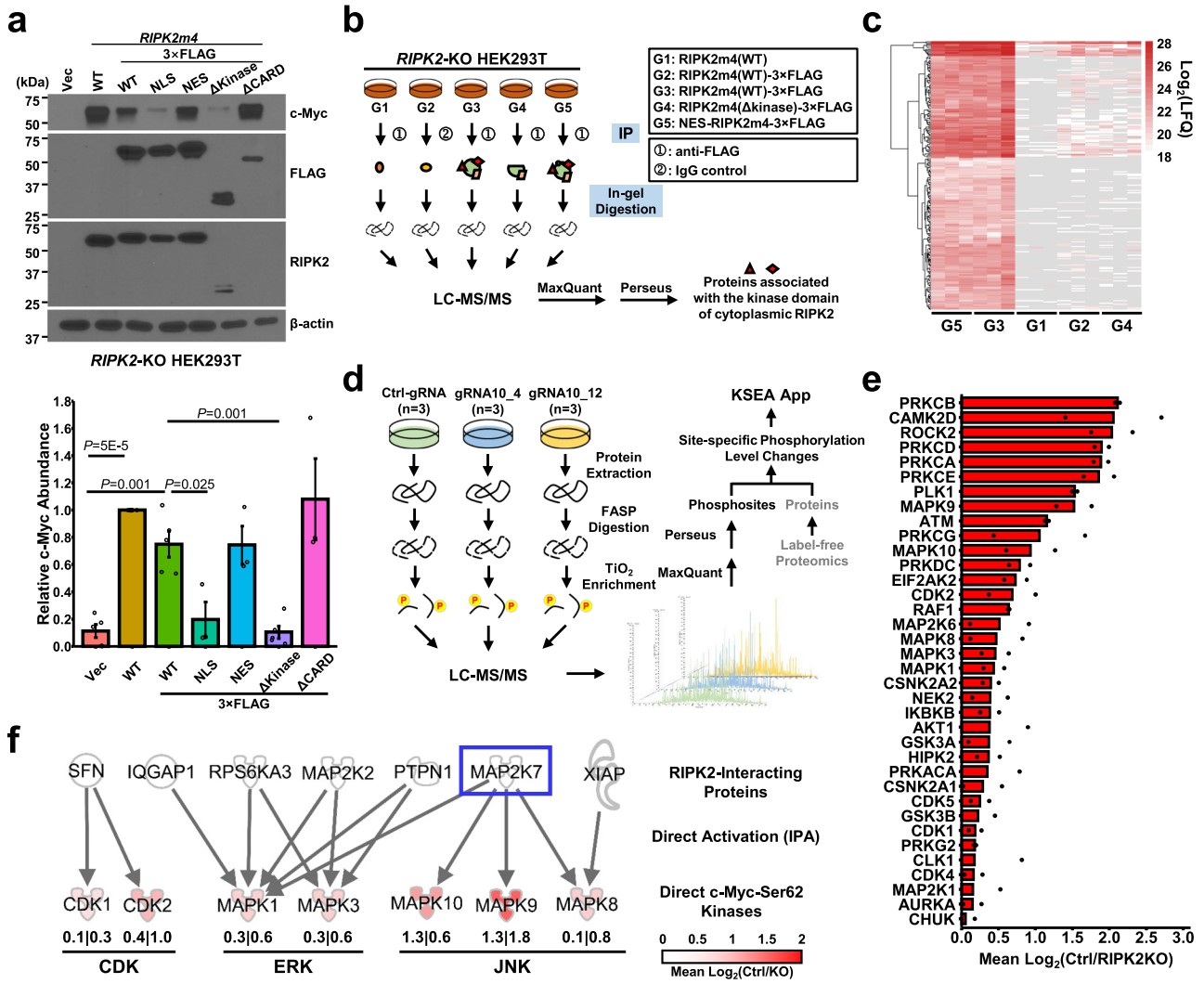

**Fig. 5 Integrative proteomics analysis identifies candidate kinase pathways mediating RIPK2 phosphorylation of c-Myc-S62. a** Representative immunoblots (upper) and quantification (lower) of c-Myc in *RIPK2*-KO HEK293T cells transiently transfected with the indicated plasmids (n = 5 biologically independent samples per group, excepting for n = 3 biologically independent samples in the ΔCARD group). Nominal *p*-values were determined by unpaired two-tailed Student's *t*-test. Data are shown as Mean ± SEM. WT: wild-type; NLS: nuclear localization signal; NES: nuclear export signal; Δkinase: deletion of the kinase domain; ΔCARD: deletion of the CARD domain. **b** Schematic of the IP-MS-based interactome analysis. *RIPK2*-KO HEK293T cells were transiently transfected with the indicated forms of plasmids (shown in the upper right box). Protein complexes were immunoprecipitated using an anti-FLAG antibody or IgG control. Immunoprecipitated protein complexes were analyzed by in-gel digestion followed by mass spectrometric analysis. Raw files were analyzed by MaxQuant and Perseus to identify proteins associated with the kinase domain of cytoplasmic RIPK2. **c** Heatmap of the 219 protein candidates specifically associated with the kinase domain of cytoplasmic RIPK2. Gray indicates zero label-free quantification (LFQ) intensities. **d** Schematic of the label-free phosphoproteomics analysis. Proteins were extracted from control or *RIPK2*-KO PC3 cells and digested into tryptic peptides using FASP (filter-aided sample preparation). Phosphopeptides were enriched by titanium dioxide (TiO₂) and analyzed by LC-MS/MS. Raw files were analyzed by MaxQuant to identify phosphopeptides, by Perseus to identify significantly differentially phosphorylated peptides, and by kinase-substrate enrichment analysis (KSEA) to identify differentially activated kinases. **e** Bar plot of the inferred activity changes of candidate kinases activated by RIPK2. For each kinase, the two dots represent Log₂Ratios in control PC3 cells relative to those in *RIPK2*-KO PC3 clones (#4 and #12), respectively. **f** Activation network connecting RIPK2-interacting protein candidates (identified by the interactome profiling) with direct c-Myc-S62 kinases downstream of RIPK2 (identified by the phosphoproteomics analysis). The numbers indicate kinase activities in control PC3 cells relative to those in *RIPK2*-KO PC3 clone#4 (left) or clone#12 (right).

were quantified based on ≥ 5 substrate phosphosites per kinase in the two *RIPK2*-KO PC3 clones, respectively (Supplementary Fig. 28d and Supplementary Data 4), with an overlap of 47 kinases (Supplementary Fig. 28e and Supplementary Data 5). RIPK2 appeared to activate 36 kinases (Fig. 5e), of which nine were reported as direct c-Myc-S62 kinases but not as direct RIPK2 substrates.

The interactome and phosphoproteomics findings raised a possibility that RIPK2 binds to and activates a protein, which in turn activates a direct c-Myc-S62 kinase. Thus, we applied IPA to

connect the 219 RIPK2-interacting proteins with the nine direct c-Myc-S62 kinases, based on whether the former can directly activate the latter. Interestingly, seven RIPK2-interacting proteins were found to be able to directly activate seven direct c-Myc-S62 kinases (Fig. 5f). Among the latter, c-Jun N-terminal kinases (JNKs) downstream of mitogen-activated protein kinase kinase 7 (MAP2K7, also known as MKK7) appeared to be most activated by RIPK2 (Fig. 5f). Together, the findings suggest that RIPK2 may indirectly phosphorylate c-Myc-S62 via multiple kinase pathways, particularly the MKK7 pathway.

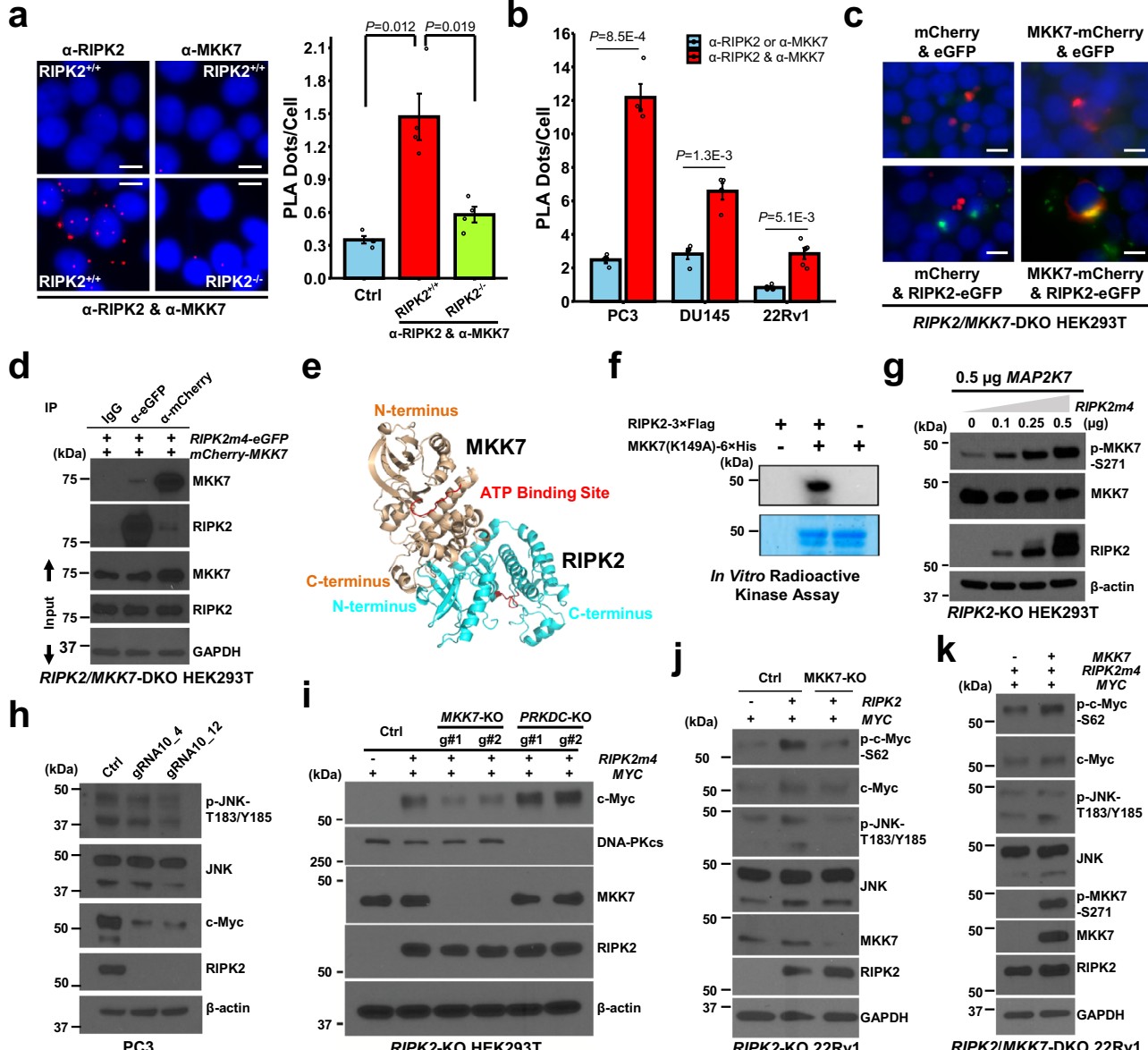

**Fig. 6 MKK7 is a major mediator of RIPK2 phosphorylation of c-Myc-S62. a** Representative images (left) and quantification (right) of PLA detecting the association of endogenous RIPK2 and MKK7 in HEK293T cells ($n = 4$ biologically independent samples per group). Scale bars, 10 μm. α-: anti-. **b** Bar plot of PLA detecting the association of endogenous RIPK2 and MKK7 in parental PC cells ($n = 4$ biologically independent samples per group). **c** Representative images of fluorescence colocalization of RIPK2 and MKK7 ectopically expressed in *RIPK2/MKK7*-DKO HEK293T cells. Scale bars, 10 μm. eGFP: enhanced green fluorescent protein. **d** Representative images of Co-IP analysis detecting the association of RIPK2 and MKK7 ectopically expressed in *RIPK2/MKK7*-DKO HEK293T cells. **e** Three-dimensional structural model of direct RIPK2 and MKK7 association in ribbon representation. **f** Representative images of radioactive in vitro RIPK2 kinase assay detecting RIPK2 phosphorylation of MKK7, using kinase-dead (K149A) MKK7 as substrate. The upper and lower panels show representative autoradiography and Coomassie blue-stained gel images, respectively. **g–k** Representative immunoblots of the indicated proteins in total lysates of **g** *RIPK2*-KO HEK293T cells transiently transfected with the indicated amounts of *RIPK2m4* plasmid, with 0.5 μg MKK7 being co-transfected to enable reliable quantification of p-MKK7, **h** control or *RIPK2*-KO PC3 cells, **i** *RIPK2*-KO HEK293T cells under the indicated conditions (g#1 and g#2: two independent gRNAs for gene knockout), **j** *RIPK2*-KO 22Rv1 cells under the indicated conditions, and **k** *RIPK2/MKK7*-DKO 22Rv1 cells under the indicated conditions. Nominal *p*-values were determined by unpaired two-tailed Student's *t*-test (**a**, **b**). Data are shown as mean ± SEM (**a**, **b**). Experiments were repeated three times independently with similar results (**c**, **d**, **f–k**).

**MKK7 is a major mediator of RIPK2 regulation of c-Myc.** To confirm the protein-protein association between RIPK2 and MKK7, we performed PLA, fluorescence colocalization, and co-IP analyses. PLA assays confirmed that endogenous RIPK2 and MKK7 are in close proximity (<40 nm) in HEK293T as well as PC3, DU145, and 22Rv1 cells under normal culture conditions, predominantly if not exclusively in the cytoplasm (Fig. 6a, b and Supplementary Fig. 29a). Fluorescence colocalization and co-IP

analyses further confirmed the colocalization and association of RIPK2 and MKK7 (Fig. 6c, d). Structural modeling suggested that the N-terminal kinase domain of RIPK2 may directly bind to the C-terminal kinase domain of MKK7 (Fig. 6e). Radioactive in vitro kinase assay showed that RIPK2 can directly phosphorylate kinase-dead (K149A) MKK7 (Fig. 6f). In addition, immunoblotting analyses showed that forced expression of RIPK2 activated MKK7 and JNK in a dose-dependent fashion (Fig. 6g and

Supplementary Fig. 29b, c) and that *RIPK2*-KO decreased the JNK phosphorylation level (a surrogate for MKK7 activity) and c-Myc protein abundance (Fig. 6h). Collectively, the results suggest that RIPK2 binds to and activates MKK7.

To validate that MKK7 is a major mediator of RIPK2 regulation of c-Myc, we knocked out *MKK7* from *RIPK2*-KO HEK293T cells with two different gRNAs by CRISPR/Cas9. Immunoblotting analysis showed that *MKK7*-KO decreased RIPK2-induced c-Myc overexpression by ~58% (Fig. 6i and Supplementary Fig. 29d), suggesting that MKK7 is a major (albeit not the only) mediator of RIPK2's regulation of c-Myc. In comparison, the knockout of *PRKDC* had no significant effect. The *PRKDC* gene encodes the DNA-dependent protein kinase catalytic subunit (DNA-PKcs), which we discovered as a RIPK2-interacting protein by IP-MS and confirmed the interaction by immunoblotting and PLA (Supplementary Fig. 30). Moreover, *MKK7*-KO substantially abrogated RIPK2-induced JNK phosphorylation and p-c-Myc-S62 upregulation (Fig. 6j and Supplementary Fig. 31a), and the effects could be rescued by ectopic expression of *MKK7* (Fig. 6k and Supplementary 31b). Together, the results indicate that MKK7 is a major mediator of RIPK2 phosphorylation of c-Myc-S62 and upregulation of c-Myc.

**MKK7 is a direct c-Myc-S62 kinase**. Unexpectedly, although both MKK7-COV-3 (a potent MKK7-selective inhibitor[53]) and JNK-IN-8 (a potent JNK-selective inhibitor[54]) completely abolished RIPK2-induced c-Jun phosphorylation (a surrogate for JNK activity), the JNK inhibitor is ~3-fold weaker than the MKK7 inhibitor in reducing RIPK2-induced p-c-Myc-S62 upregulation (Fig. 7a and Supplementary Fig. 32a, b). In addition, higher concentrations (up to 25 μM) of JNK-IN-8 could not enhance its inhibition of RIPK2-induced c-Myc-S62 phosphorylation (Supplementary Fig. 32c). These findings raised a possibility that MKK7 can phosphorylate c-Myc both directly (main route) and via JNK (minor route).

In support of the hypothesis, PLA and immunofluorescence analyses showed that MKK7 colocalizes with c-Myc, predominantly (if not exclusively) in nuclei (Fig. 7b, c and Supplementary Fig. 33). In vitro kinase assays showed that recombinant MKK7 could directly phosphorylate c-Myc-S62 (Fig. 7d) and that immunoprecipitated constitutively active MKK7-3E (S271E, T275E, S277E) is much more efficient than immunoprecipitated kinase-dead MKK7-K149A in phosphorylating c-Myc-S62 (Supplementary Fig. S34). The residual c-Myc-S62 phosphorylation under the K149A condition is likely due to the co-immunoprecipitation of other direct c-Myc-S62 kinases or the residual kinase activity of MKK7-K149A. Moreover, in the absence of RIPK2, forced expression of constitutively active MKK7 increased c-Myc and p-c-Myc-S62 levels, compared with wild-type and kinase-dead MKK7 (Fig. 7e). Together, the results suggest that active MKK7 can directly phosphorylate c-Myc in nuclei. To the best of our knowledge, this is the first report that MKK7 is a direct c-Myc-S62 kinase.

According to the Cell Atlas, MKK7 is predominantly localized in the cytoplasm of HEK293T and PC3 cells. However, our subcellular fractionation analysis showed that, when RIPK2 was ectopically expressed, active and total MKK7 were primarily localized in the nuclei, whereas active and total JNK were predominantly localized in the cytoplasm (Fig. 7f). In addition, PLA analysis showed that forced expression of RIPK2 increased the amount of the MKK7-c-Myc complex in the nuclei (Fig. 7g). Given that RIPK2 binds to and activates MKK7 in the cytoplasm (Fig. 6) and that the activation of certain kinases may stimulate their nuclear translocation[55], the results support a model in which RIPK2 activation of MKK7 in the cytoplasm stimulates

translocation of MKK7 into nuclei, where MKK7 phosphorylates and stabilizes c-Myc mainly directly and only partially via JNK.

**Pharmacological inhibition of RIPK2 inactivates RIPK2/ MKK7/c-Myc signaling and suppresses PC metastatic outgrowth**. According to cell-free biochemical assays, RIPK2 can be potently inhibited by multiple small-molecule inhibitors such as GSK583 and ponatinib (Supplementary Table 1)[56,57]. However, the cellular potency of a compound can be substantially lower than the biochemical potency to varying degrees (up to several orders of magnitude)[58]. Our cell-based assay showed that under regular cell culture conditions containing 10% fetal bovine serum (FBS), 2 h treatment with 10 μM GSK583 or 5 μM ponatinib substantially decreased RIPK2-induced MKK7 phosphorylation to a similar extent (Fig. 8a). In addition, 5 μM ponatinib is more potent than 10 μM GSK583 in inhibiting RIPK2-induced JNK activation (Fig. 8b and Supplementary Fig. 35a) and in decreasing endogenous p-JNK, p-c-Myc-S62, and c-Myc levels in parental PC cells (Fig. 8c and Supplementary Fig. 35b). Taken together, the results suggest that both GSK583 and ponatinib can inactivate cellular RIPK2/MKK7(/JNK)/c-Myc signaling, and ponatinib is more potent than GSK583 in this aspect.

To determine whether pharmacological inhibition of RIPK2 reduces the metastatic potential of PC cells, we performed in vitro 3D spheroid invasion and colony formation assays. Both GSK583 and ponatinib significantly inhibited the spheroid invasion of PC3 cells, and ponatinib was more potent than GSK583 (Fig. 8d and Supplementary Fig. 36a). GSK583 significantly inhibited colony formation of PC3, DU145, and 22Rv1 cells by ≥40% at a high dose (10 μM) but not at low doses (≤1 μM) (Fig. 8e and Supplementary Fig. 36b, c). In comparison, ponatinib more potently inhibited the colony formation of PC cells, with $IC_{50}$ values of <1 μM (Fig. 8f and Supplementary Fig. 36d, e). Notably, 10 μM GSK583 and 1 μM ponatinib only had marginal effects on PC cell viability (Supplementary Fig. 37). Collectively, the in vitro assays suggested that pharmacological inhibition of RIPK2 can effectively attenuate the metastatic potential of PC cells.

To investigate whether pharmacological inhibition of RIPK2 can suppress PC metastasis in a setting that mimics the clinical scenario, we injected the aforementioned luciferase-tagged control 22Rv1 cells intracardially into male SCID/Beige mice and waited for 9 days to allow for the formation of early metastatic lesions, prior to drug treatment (Fig. 8g). To determine whether GSK583 impairs metastatic outgrowth, we randomized the mice into two groups (Supplementary Fig. 38a), followed by daily treatment with vehicle control or GSK583 (10 mg/kg/day). Phenocopying the *RIPK2*-KO result (Fig. 2h), GSK583 significantly suppressed the metastatic progression of 22Rv1 cells in vivo (50% reduction at week 4) (Fig. 8h, i and Supplementary Fig. 38b), without significant effect on mouse weight (Supplementary Fig. 38c), lethargy, or loss of appetite. Compared with the whole body, GSK583 more significantly decreased the bioluminescence imaging (BLI) intensities (by 67% at week 4) in the ventral-side mid-body, which largely (if not exclusively) correspond to liver metastasis as supported by histopathological analysis (Supplementary Fig. 39). Notably, studies have shown that, among common visceral metastases, PC patients with liver metastases exhibited the worst median overall survival[59]. Taken together, the results suggest that pharmacologically targeting RIPK2 is a viable strategy to inhibit PC metastasis.

Our pilot study showed that a higher dose (30 mg/kg/d) of GSK583 was toxic to mice (data not shown). Thus, we asked whether the FDA-approved ponatinib, a more potent RIPK2/MKK7/c-Myc pathway inhibitor than GSK583 and two other commercially available RIPK2-selective inhibitors (Supplementary Fig. 40), can

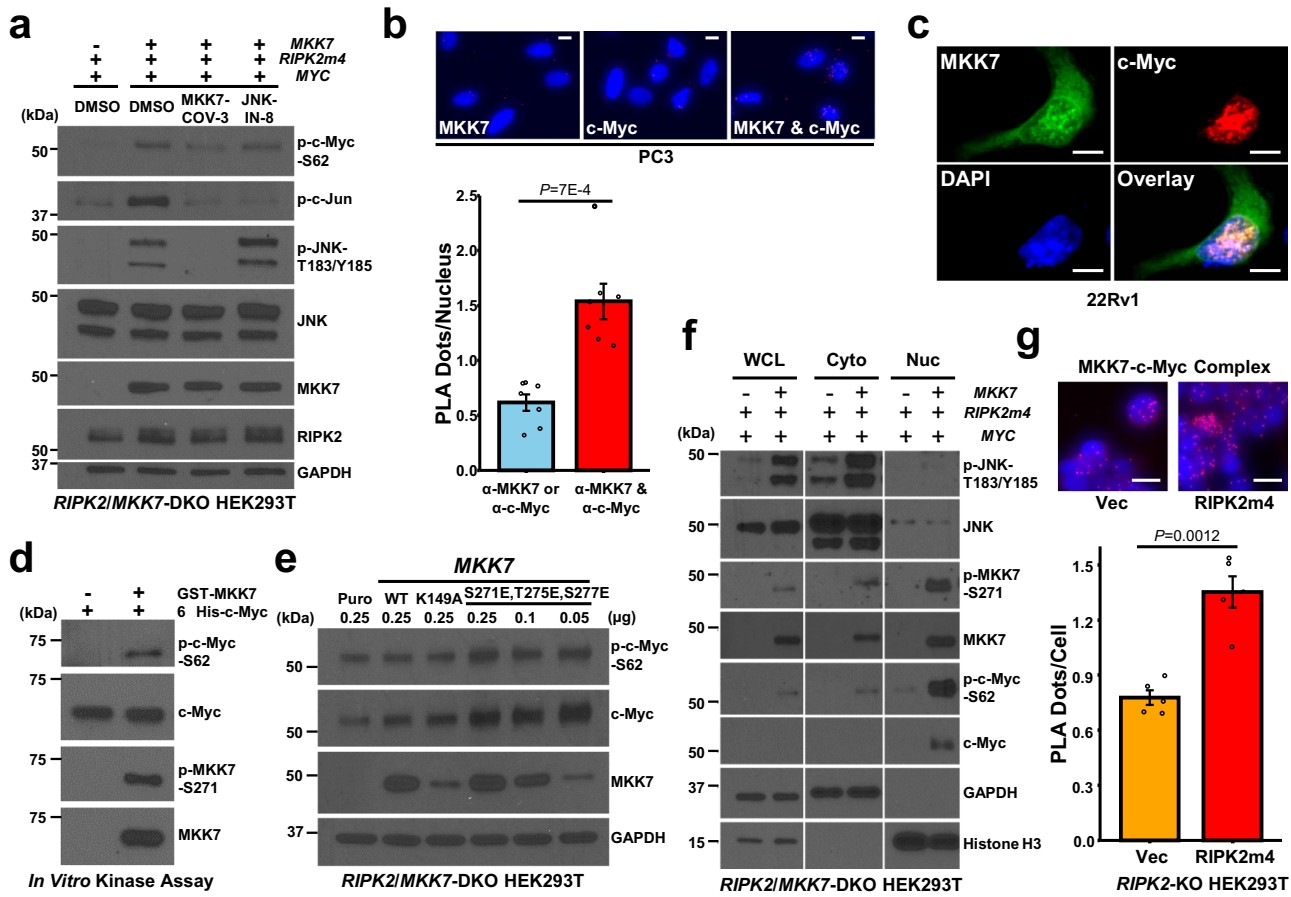

**Fig. 7 MKK7 is a direct c-Myc-S62 kinase. a** Representative immunoblots of the indicated proteins in total lysates of *RIPK2/MKK7*-DKO HEK293T cells, which were transiently co-transfected with 0.5 μg *MYC*, 0.5 μg *RIPK2m4*, and 0.1 μg *MAP2K7* or vector plasmids and then treated with vehicle control, MKK7-COV-3 (25 μM), or JNK-IN-8 (1 μM) in 10% FBS-containing medium for 2 h. **b** Representative images (upper) and quantification (lower) of PLA detecting the association of endogenous MKK7 and c-Myc in PC3 cells (*n* = 7 biologically independent samples per group). Scale bars, 10 μm. **c** Representative images of immunofluorescence colocalization of MKK7 and c-Myc in *RIPK2/MKK7*-DKO 22Rv1 cells, which were transiently co-transfected with *RIPK2m4*, *MKK7-His*, and *MYC* plasmids. Scale bars, 10 μm. **d** Representative immunoblots of the indicated proteins after in vitro kinase reactions. **e** Representative immunoblots of the indicated proteins in total lysates of *RIPK2/MKK7*-DKO HEK293T cells, which were transiently transfected with the indicated forms of *MKK7* plasmids or vector control. **f** Representative immunoblots of the indicated proteins in total lysates, cytoplasmic fraction, and nuclear fraction of *RIPK2/MKK7*-DKO HEK293T cells, which were transiently co-transfected with *RIPK2m4*, *MYC*, and *MKK7* or vector control. WCL: whole-cell lysate; Cyto: cytoplasmic; Nuc: nuclear. **g** Representative images (upper) and quantification (lower) of PLA detecting the MKK7-c-Myc complex in *RIPK2*-KO HEK293T cells, which were transiently transfected with *MYC* and *RIPK2m4* or vector control (*n* = 5 biologically independent samples per group). Scale bars, 20 μm. Nominal *p*-values were determined by unpaired two-tailed Student's *t*-test (**b**, **g**). Data are Mean ± SEM (**b**, **g**). Experiments were repeated three times (**a**, **d**) or twice (**e**, **f**) independently with similar results.

more effectively inhibit metastatic outgrowth. Indeed, ponatinib substantially reduced 22Rv1 metastatic outgrowth in a dose-dependent fashion and did not cause discernable toxicity when used at ≤15 mg/kg/day (Fig. 8j and Supplementary Fig. 41). At week 4, low-dose and high-dose ponatinib reduced total BLI intensities by 74% and 92%, respectively (Fig. 8j, k). Of note, the daily (low) dose of 6 mg/kg ponatinib in mice is estimated to be equivalent to 30 mg for a 60 kg patient[60], a commonly used and tolerated dose for clinical treatment. Collectively, the findings suggest that the FDA-approved ponatinib can effectively inhibit the outgrowth of PC metastases and thus holds great potential for repurposing as an agent to suppress metastasis therapeutically.

## Discussion

Metastasis is the major cause of PC morbidity and mortality. When cancer cells metastasize to distant organs and grow into overt metastases, even if they initially respond to therapies, they almost invariably develop therapeutic resistance. Thus,

preventing or delaying metastasis has increasingly been appreciated as an effective strategy to improve the quality of life and prolong the survival time of cancer patients[61].

In this study, via integrating multi-omics and functional analyses, we discovered and established RIPK2 as a clinically actionable target for inhibiting PC metastasis. Phenotypically, both genetic and pharmacological inhibition of RIPK2 suppressed PC cell invasion and colony formation in vitro and metastasis in vivo. Mechanistically, RIPK2 phosphorylates, stabilizes, and activates c-Myc, largely by activating the MKK7/c-Myc and MKK7/JNK/c-Myc phosphorylation cascades. Importantly, phosphorylation cascade is a highly effective strategy for signal amplification, which may explain why *RIPK2* and *MYC* activity scores are strongly correlated in PC and many other cancer types. To the best of our knowledge, this study represents the first establishment of the functional relationship between three well-studied proteins—the proinflammatory kinase RIPK2, the MAPK kinase MKK7, and the oncoprotein c-Myc. Moreover, these mechanistic findings are clinically relevant because (1) *RIPK2* and

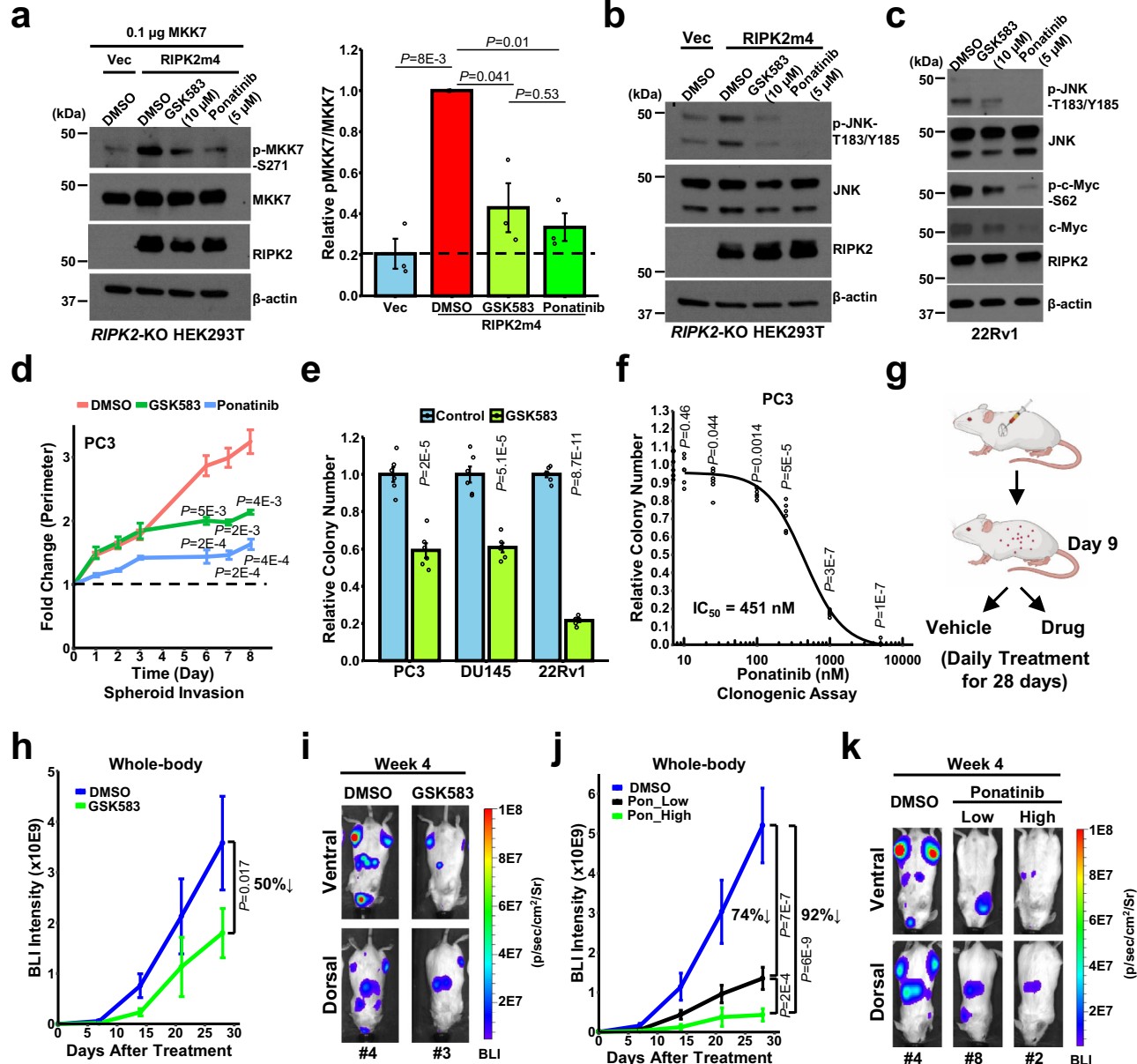

**Fig. 8 Pharmacological inhibition of RIPK2 inactivates RIPK2/MKK7/c-Myc signaling and suppresses PC metastatic outgrowth. a** Representative immunoblots (left) and quantification (right) of p-MKK7-S271 in total lysates of *RIPK2*-KO HEK293T cells under the indicated conditions (2 h treatment in the presence of 10% FBS) (*n* = 3 biologically independent samples per group). **b, c** Representative immunoblots of the indicated proteins in total lysates of **b** *RIPK2*-KO HEK293T cells or **c** 22Rv1 cells under the indicated conditions (2 h treatment). Experiments were repeated three times independently with similar results (**b, c**). **d** Quantification of 3D spheroid invasion assays of PC3 cells treated with vehicle control, 10 μM GSK583, or 5 μM ponatinib (*n* = 6 biologically independent samples per group). **e** Bar plot of anchorage-dependent colony formation of PC3, DU145, and 22Rv1 cells treated with vehicle control or 10 μM GSK583 (*n* = 6 biologically independent samples per group). **f** Drug response curve of anchorage-dependent colony formation of PC3 cells treated with vehicle control or the indicated concentrations of ponatinib (*n* = 6 biologically independent samples per group). **g** Schematic diagram of drug treatment for the inhibition of PC metastasis in vivo. Created with BioRender.com. **h** Effect of GSK583 treatment (10 mg/kg/day) on total BLI intensities in biologically independent mice (vehicle, *n* = 5; GSK583, *n* = 7), starting on day 9 following the i.c. injection. **i** Representative BLI images of the ventral (upper) and dorsal (lower) sides of mice treated with vehicle control or GSK583 for 4 weeks. **j** Effect of low-dose (6 mg/kg/day) or high dose (30 mg/kg/day for 10 days followed by 15 mg/kg/day for 18 days) ponatinib treatment on total BLI intensities in biologically independent mice (vehicle, *n* = 6; low-dose ponatinib, *n* = 8; high-dose ponatinib, *n* = 7), starting on day 9 following the i.c. injection. Pon: ponatinib. **k** Representative BLI images of the ventral (upper) and dorsal (lower) sides of mice treated with vehicle control, low-dose ponatinib, or high-dose ponatinib for 4 weeks. Nominal *p*-values were determined by unpaired two-tailed Student's *t*-test (**a, d–f**) or two-way ANOVA (**h, j**). Data are Mean ± SEM (**a, d–f, h, j**).

*MYC* are frequently co-amplified/gained in lethal PC and several other cancer types, and (2) *RIPK2* and *MYC* activity scores are strongly correlated in clinical tissue specimens of 32 cancer types. These results also have high translational value because RIPK2 can be inhibited by ponatinib (Iclusig)—an FDA-approved agent

whose safety profiles are known. When being administered at 30 mg orally once daily, ponatinib is tolerated and suitable for such a metastasis-inhibition strategy. The benefit-risk profile of ponatinib might be further improved by initiating treatment at 45 mg daily and then reducing the dose to 15 mg daily, a strategy

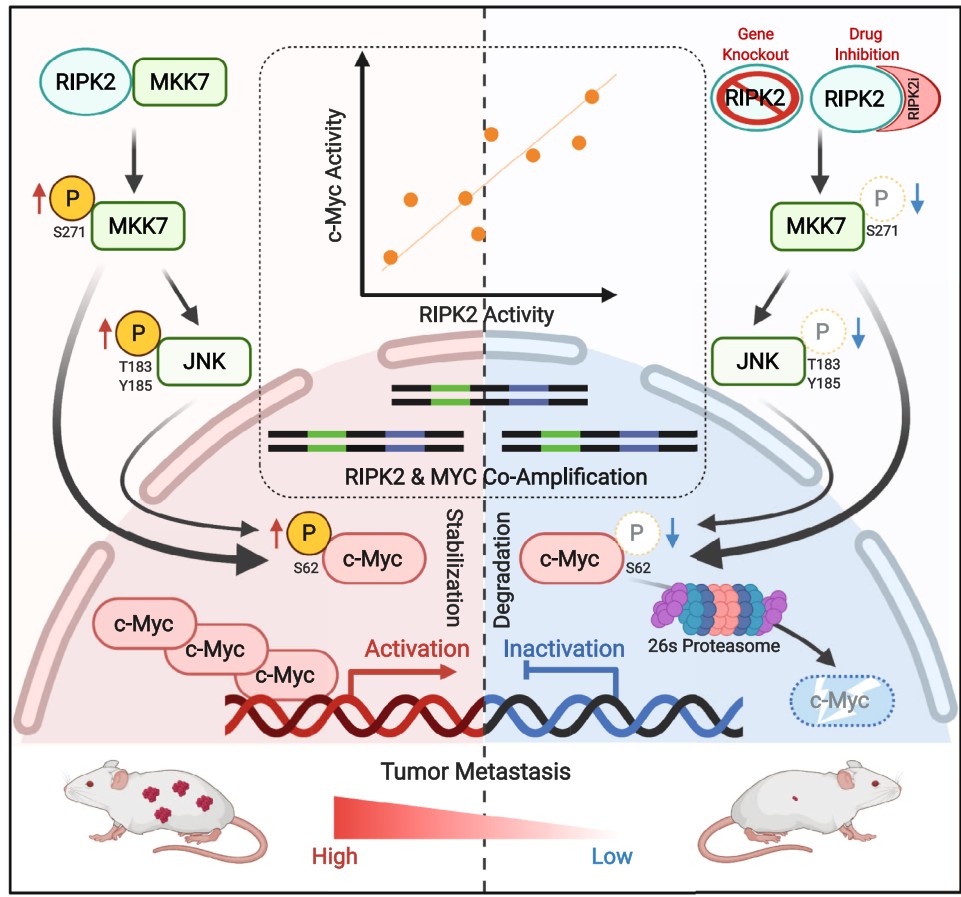

**Fig. 9 Schematic summary of the findings in this study.** RIPK2 binds to MKK7 and activates the MKK7(/JNK)/c-Myc phosphorylation cascades, thereby stabilizing the c-Myc oncoprotein and enhancing c-Myc activity, which is required for PC metastasis. The non-canonical RIPK2/MKK7/c-Myc signaling pathways can be blocked by genetic or pharmacological inhibition of RIPK2, resulting in the proteasomal degradation of c-Myc and the inhibition of c-Myc activity and PC metastasis. Of clinical relevance, *RIPK2* and *MYC* are frequently co-amplified/gained in PC and several other cancer types, and their activity scores are strongly correlated across many cancers. Created with BioRender.com.

applied in the OPTIC trial (NCT02467270). Conversely, directly targeting c-Myc in human patients remains elusive.

Although RIPK2 has been studied for over two decades, its direct substrate proteins remain poorly defined. Our interactome and phosphoproteomics data sets provide a rich source for discovering direct RIPK2 substrates. In particular, our comprehensive interactome analysis identified a large number (>200) of RIPK2-interacting proteins, of which the vast majority have not been reported (Supplementary Data 3). Many RIPK2-associated proteins in the canonical RIPK2 pathway, such as NOD1/2, TAB2/3, and IKK, were not identified by our interactome analysis. This is probably because our in vitro cell culture system does not contain bacteria-released ligands (e.g., muramyl dipeptide), which are required for the activation of the canonical pathway. In addition, according to the Cancer Cell Line Encyclopedia (CCLE) database, the expression levels of NOD1 and NOD2 in PC cell lines are generally very low. Nevertheless, these are based on in vitro cell culture models, so the importance of the canonical NOD/RIPK2 pathway in PC patients cannot yet be ruled out. In this study, we discovered that RIPK2 interacts with six protein kinases: MKK7 (*MAP2K7*), DNA-PKcs (*PRKDC*), RSK2 (*RPS6KA3*), MST4 (*STK26*), MEK2 (*MAP2K2*), and CSK (*CSK*), many of which were implicated in cancer metastasis[62]. Delineating the hierarchy between these kinases and RIPK2 will provide further insights into this protein functions in cancer metastasis.

In summary, we identified RIPK2 as an actionable drug target for inhibiting PC metastasis, operating largely via the noncanonical RIPK2/MKK7(/JNK)/c-Myc signaling pathway discovered in this study. Targeting RIPK2 by genetic knockout or small-molecule inhibitors effectively inactivates this pathway, leading to the destabilization of the c-Myc oncoprotein and the inhibition of metastasis (Fig. 9). Besides PC, *RIPK2* is frequently genetically altered in several other cancers, its overexpression is associated with shorter overall survival in nine cancer types, and its activity scores are highly correlated with *MYC* activity scores across 32 cancer types. Thus, clinical trials targeting RIPK2— either alone or in combination with established or emerging therapies—are warranted for personalized anti-metastatic therapy towards substantially improving clinical outcomes.

## Methods

**Public resources**. Publicly accessible web portals, such as the cBioPortal[23], the PCTA[24], and the Pharos for human druggable genome[25], were used to analyze clinical omics data sets. The cBioPortal (cbioportal.org) hosts omics data from large consortium efforts and publications from individual labs. In the cBioPortal environment, the copy number alterations in three mCRPC cohorts (Grasso et al., $n = 61$; Abida et al., $n = 444$; and Robinson et al., $n = 150$)[26–28] were analyzed to calculate the alteration frequencies, and the TCGA PanCancer Atlas transcriptomics profiles were analyzed for Kaplan–Meier survival analysis and to compute RIPK2 and MYC activity $Z$ scores. The PCTA portal (thepcta.org) contains the PCTA dataset[24], which comprises transcriptomic data from 38 PC cohorts, as well as the TCGA Firehose Legacy dataset[63], which contains transcriptomic profiles from 499 PC specimens. The TCGA PanCancer Atlas and Firehose Legacy data sets are quite similar but the former was batch-corrected[63]. In the PCTA environment, the "Expression View" function was used to visualize the expression trends of genes or gene lists and the expression levels of RIPK2 in different PCS and PAM50 subtypes, while the "Correlation View" function was

used to calculate the correlation coefficients between gene expression levels and MYC activity $Z$ scores and between RIPK2 and MYC activity $Z$ scores. The Pharos (pharos.nih.gov) is a comprehensive and integrated knowledge base for the Druggable Genome program. It was used to retrieve genes for which clinical-grade ($T_{clin}$) and chemical-grade ($T_{chem}$, with $IC_{50} < 30$ nM) inhibitors have already been developed.

**Generation of mutant and fusion cDNA.** Primer sequences for molecular cloning were listed in Supplementary Table 2. Site-directed mutagenesis was conducted with the QuickChange II XL Site-Directed Mutagenesis Kit (Agilent #200521). RIPK2m4 was generated by synonymous substitution of 648 A->T, 651 T->A, 654 C->A, and 657 A->C. The following fusion cDNAs were generated by PCR and cloned into the pLenti-puro vector: 3 × FLAG-tag was fused to the C terminus of target genes; NLS and NES to the N-terminus of RIPK2m4; 6 × His-tag to the C terminus of MKK7. The RIPK2m4 sequence was cloned into the pLJM-eGFP vector to form RIPK2m4-eGFP. The MKK7 sequence was cloned into the mCherry2-N1 vector to form MKK7-mCherry. All plasmids were validated by sequencing (Laragen).

**RNA extraction and quantitative real-time polymerase chain reaction (RT-PCR).** Total RNA was extracted from cells using TRIZOL (Invitrogen #15596026). Messenger RNA was converted to the first-strand cDNA using iScript cDNA synthesis kit (Bio-Rad #1708891), followed by RT-PCR reaction using SYBR Green PCR Master Kit in QuantStudio 5 Systems (Applied Biosystems #4309155). Gene expression was normalized to GAPDH using the comparative CT method.

**Cell lines.** PC3 (#CRL-1435), DU145 (#HTB-81), 22Rv1 (#CRL-2505), LNCaP (#CRL-1740), RWPE-1 (#CRL-11609), RWPE-2 (#CRL-11610), and HEK293T (#CRL-11268) cells were obtained from the American Type Culture Collection (ATCC). The cell lines were authenticated using the Promega PowerPlex 16 system DNA typing (Laragen). Mycoplasma contamination was routinely monitored using the MycoAlert PLUS Mycoplasma Detection Kit (Lonza #LT07-118).

**Gene knockout by CRISPR/Cas9.** The sequences and targeting domains of sgRNAs are shown in Supplementary Table 3. All guide RNAs were cloned into LentiGuide-Puro (Addgene #52963) followed by sequencing validation (Laragen). Transfer (LentiGuide-Puro or LentiCas9-Blast, Addgene #52962), packing (psPAX2, Addgene #12260), and envelope (pMD2.G, Addgene #12259) plasmids were co-transfected into HEK293T cells to produce lentivirus containing sgRNAs and Cas9. To generate stable cell lines with gene knockout, cells were infected with lentivirus of Cas9 and sgRNA supplemented with 10 μg/mL polybrene.

**Transient transfection.** When cells grew to a confluency of ~50%, a specified amount of plasmid was transfected into cells using Turbofectin 8.0 (Origene #TF81001) by following the manufacturer's instructions. In the RIPK2 and c-Myc co-transfection experiments, 0.5 μg per plasmid was used unless otherwise indicated. In some cell line experiments, cells were co-transfected with the *MYC* or *MKK7* plasmid to enable reliable quantification of p-c-Myc-62 or p-MKK-S271, respectively.

**In vitro cell assays.** Cell proliferation assay was performed using the Cell Counting Kit-8 (CCK-8) (Sigma Aldrich #96992) according to the manufacturer's instructions. Migration and invasion assays were performed essentially as described[64]. For migration assay, the outside membrane of Transwell inserts (8 μm pore size) was coated with 50 μL of 15 μg/mL Collagen Type I. A total of $5 \times 10^4$ control or *RIPK2*-KO PC3 cells in 200 μL serum-free medium were seeded in the upper chamber, followed by adding 600 μL 10% FBS-containing medium to the lower chamber. After incubation at 37 °C for 6 h, cells were fixed by 4% paraformaldehyde and stained with 0.12 mg/mL crystal violet solution for 15 min. Cells on the inside membrane of Boyden chambers were removed by cotton swabs. Migrated cells were imaged under an All-in-one Keyence microscope, and total area (μm$^2$) of migrated cells was automatically calculated using the Keyence image analyzer. For the invasion assay, the inside membrane of Transwell inserts (8 μm pore size) was coated with a 50 μL Matrigel matrix (1:50 diluted). For each sample, 200 μL ($1 \times 10^6$ cells/mL) of control or *RIPK2*-KO PC3, DU145, or 22Rv1 cells suspended in serum-free medium were seeded in the upper chamber. 600 μL of 10% FBS-containing medium was added to the lower chamber. After incubation at 37 °C for 24 h, cells were fixed, stained, cleaned, imaged, and quantified as described in the migration assay. 3D spheroid invasion assays were performed as described[65]. Briefly, 4000 PC3 cells in 200 μL DMEM medium supplemented with 10% FBS, 2 mM L-glutamine, 100 units/mL penicillin, and 100 μg/mL streptomycin were plated in Ultra-Low attachment U-bottom 96-well plates (Corning, #4515) and cultured for four days to form tumor spheroids. 100 μL Matrigel was gently dispensed to each well containing a tumor spheroid and incubated at 37 °C for 1 h. For the comparison of control and *RIPK2*-KO PC3 spheroids, 100 μL DMEM medium was added to each well. For the drug treatment experiment, 100 μL DMEM medium supplemented with DMSO, 10 μM GSK583, or 5 μM Ponatinib was added. Images at indicated time points were analyzed by the INSIDIA macro

in the environment of Fiji. Relative perimeter reflects the complexity of the invading edge and can provide a quick and easy analysis of spheroid invasion[66]; thus, it was used to quantify 3D invasion. Clonogenic (anchorage-dependent colony formation) and soft agar (anchorage-independent colony formation) assays were performed as described[67,68]. For clonogenic assays, 2 mL of PC3 (250/mL), DU145 (250/mL), or 22Rv1 (1000/mL) cells were seeded in 6-well plates and cultured in 10% FBS-containing media, which were replaced every 2–3 days. After 10–14 days, cells were fixed in 4% paraformaldehyde for 15 min and rinsed with DPBS three times. Cells were stained with 120 μg/mL crystal violet solution. For soft agar assays, 1.5 mL of 0.5% Nobel agar solution was plated in each well of 6-well plates as the bottom layer. 1.5 mL of $6.67 \times 10^3$/mL (for PC3 and DU145) or $1.33 \times 10^4$/mL (for 22Rv1) cells in 0.3% noble agar solution were plated in 6-well plates as the top layer. Cells were cultured in DMEM supplemented with 10% FBS, 2 mM L-glutamine, 100 units/mL penicillin and 100 μg/mL streptomycin. The medium was replaced every 2–3 days. After 2–3 weeks, cell colonies were stained with 0.1% crystal violet solution for 20 minutes and washed with deionized water five times. For both clonogenic and soft agar assays, cell colonies were quantified under an All-in-one microscope (Keyence) as described in the migration assay.

**Animal studies.** All experimental protocols and procedures were approved by the Institutional Animal Care and Use Committee (IACUC) at Cedars-Sinai Medical Center and the Animal Care and Use Review Office at the Department of Defense. All relevant ethical regulations, standards, and norms were rigorously adhered to. Mice were housed at 74 °F (±2 °F) with ambient humidity. The light cycle of animal rooms was 10 h of light and 14 h of dark.

*Subcutaneous xenograft model.* For In vivo tumor growth assays, control or *RIPK2*-KO 22Rv1 cells were adjusted to $1.5 \times 10^7$ cells/mL in DPBS, followed by mixing with Matrigel at a 1:1 ratio (v/v). For each male SCID/Beige mouse (7-weeks-old; Charles River #CRL:250; CB17.Cg-*Prkdc$^{scid}$Lyst$^{bg-J}$*/Crl), 100 μL mixture was subcutaneously injected into both flanks. Tumor length and width were measured with a caliper and tumor volumes were calculated using the formula of (length × width$^2$)/2. At the endpoint, mice were euthanized and tumor xenografts were collected and weighted.

*Experimental metastasis model.* For in vivo metastasis assays, luciferase-expressing cells were sorted with a FACSAria III (FACSDiva software (v8.0.1), BD Biosciences) using the same gate for GFP. Control and *RIPK2*-KO 22Rv1 cells expressing similar levels of luciferase were selected and expanded. To each male SCID/Beige mouse (7-weeks-old; Charles River #CRL:250; CB17.Cg-*Prkdc$^{sci-d}$Lyst$^{bg-J}$*/Crl), 100 μL of $1 \times 10^7$ cells/mL in DPBS was intracardially injected into the left cardiac ventricle. Each week, 150 μL of 30 mg/mL D-luciferin was intraperitoneally injected into each mouse, followed by measuring tumor metastasis with an IVIS Spectrum In Vivo Imaging System (PerkinElmer). Luciferase activity was quantified by the Living Image software (Caliper Life Sciences, v4.3.1).

*Established metastasis model.* For drug treatment, to each male SCID/Beige mouse (8-week-old; Charles River #CRL:250; CB17.Cg-*Prkdc$^{scid}$Lyst$^{bg-J}$*/Crl), 100 μL of $2.5 \times 10^6$ cells/mL in DPBS was intracardially injected. Nine days later, mice were randomized into two groups (DMSO vs. GSK583) or three groups (DMSO vs. low-dose ponatinib vs. high-dose ponatinib, LC Laboratories #P-7022). Two more mice were assigned to drug treatment groups in preparation for possible drug toxicity deaths, compared with the control group. Mice were daily administered by oral gavage (po qd) with a drug or vehicle control for 28 days. Metastases were visualized by bioluminescence imaging every week as described above. At the endpoint, mice were sacrificed by standard necropsy, and metastases were subjected to histopathology analyses.

**Hematoxylin and eosin (H&E) staining and immunohistochemistry (IHC).** H&E and IHC were carried out by the Cedars-Sinai Biobank & Translational Pathology core by following standardized protocols. For IHC staining of tissue microarrays, the anti-RIPK2 antibody (Sigma Aldrich #HPA015273) was used at 1:100 dilution. Stained slides were digitized using Aperio AT Turbo (Leica Biosystems). Cancer areas and normal glands were annotated by an expert pathologist (X.Y.) in the H&E images. The annotations were digitally transferred onto the IHC images, which were exported for image analysis in the Leica Tissue IA software package (Leica Biosystems). Protein expression was quantified by the mean 3,3'-diaminobenzidine (DAB) staining intensity of pixels in the annotated normal and tumor areas. DAB staining was automatically deconvolved from hematoxylin by the software.

**c-Myc luciferase assay.** The c-Myc activity was assessed using the Myc Reporter kit (BPS Biosciences #60519) and the Dual-Luciferase Reporter System (Promega #E1910) according to the manufacturers' instructions. c-Myc activities were determined by the ratios of firefly to Renilla luciferase activities.

**Label-free proteomics.** Label-free proteomics was performed essentially as described[69]. Briefly, for each cell clone, three biological replicates of cultured cells

($\sim 5 \times 10^6$) were harvested, and cell pellets were lysed with about three volumes of lysis buffer (80 mM Tris-HCl, 4% SDS, 100 mM DTT, pH7.4). Cell lysates were sonicated to reduce the viscosity, and protein concentrations were measured using the Pierce 660 nm Protein Assay Kit. For each replicate, 50 μg protein lysate was alkylated by 55 mM iodoacetamide and digested with trypsin (Promega, #V5280) in Microcon-YM30 spin filters (Millipore), using the filter-aid sample preparation (FASP) method. Each tryptic peptide sample was analyzed by liquid chromatography-tandem mass spectrometry (LC-MS/MS) twice. About 1 μg peptide was separated on a 50-cm EASY-Spray column using a 200-min LC gradient at the flow rate of 150 nL/min. Separated peptides were analyzed with an LTQ Orbitrap Elite (Thermo Scientific) in a data-dependent manner with the Xcalibur (v4.1) operating system. The acquired MS data (24 RAW files) were searched against the Uniprot_Human database (released on 01/22/2016, containing 20,985 sequences) using MaxQuant (v1.5.5.1). A stringent cutoff of 1% FDR was applied to filter the identifications of peptide-spectrum matches, peptide, and protein groups. The mass spectrometry proteomics data have been deposited to the ProteomeXchange Consortium (http://proteomecentral.proteomexchange.org) via the PRIDE[70] partner repository with the database identifier PXD018890, where more detailed experimental information was included. Perseus (v1.6.6.0) was applied to perform quality assessment and statistical analysis. Proteins identified from the reverse decoy and contaminating protein sequence databases as well as those with site-only identifications were removed. For statistical comparison, all LFQ intensity values were log2-transformed, and only proteins with ≥3 valid values in each group were used. Unpaired two-tailed Welch's t-test followed by Benjamini–Hochberg adjustment was used to calculate p and q values, respectively. Protein groups meeting the criteria of $p < 0.05$ and log2-transformed fold change of >0.5 in absolute value were considered as significantly changed proteins for overlapping analysis. To calculate combined p and q values, Stouffer's method followed by Benjamini–Hochberg adjustment was applied.

**Interactome analysis by immunoprecipitation-mass spectrometry (IP-MS).**
After cell transfection with indicated plasmids for 48 h, cells were harvested and lysed. 500 μL of 2 mg/mL protein solution was pre-cleared by incubating with 60 μL of 50% immobilized protein A/G Plus agarose bead slurry (Thermo Scientific #20423) for 2 h at 4 °C. The pre-cleared protein solution was incubated with 3 μg anti-FLAG antibody (Sigma Aldrich #F1804) or IgG (Millipore #12–371) overnight at 4 °C. The next day, 60 μL of 50% immobilized protein A/G gel slurry was added to each sample, followed by incubation on a vertical shaker for 2 h at 4 °C. After washing, bound proteins were eluted and analyzed by gel-enhanced liquid chromatography-tandem mass spectrometry (GeLC-MS/MS) essentially as described[69,71]. Tryptic peptides were analyzed using an EASY-nLC 1200 connected to an Orbitrap Fusion Lumos (Thermo Scientific) operated in a data-dependent manner. The acquired MS data (20 RAW files) were searched against the Uniprot_Human database (released on 03/30/2018, containing 93,316 sequences) using MaxQuant (v1.5.5.1). The data have been deposited to the PRIDE (identifier: PXD018870). Perseus (v1.6.6.0) was applied to perform quality assessment and statistical analysis as described in the label-free proteomics analysis. Missing data were imputed from normal distribution by Perseus, using the default values (width 0.3; down shift 1.8). The p and q values were computed as described above. Protein groups meeting the criteria of $q < 0.05$ and log2-transformed fold change of > 2 were accepted as significantly enriched proteins. To identify protein candidates that bind to the kinase domain but no other regions of RIPK2, two criteria were applied: (1) proteins are significantly enriched in both experimental groups (i.e., G3 and G5) compared with the three control groups (i.e., G1, G2, and G4), and (2) proteins are not significantly enriched in the control group G4 compared with the other control groups G1 and G2.

**Phosphoproteomics.** From regularly cultured control and RIPK2-KO PC3 cells (described in the label-free proteomics section), 1 mg protein was reduced, alkylated, and digested with trypsin in Amicon Ultra-4 centrifugal filter units (Millipore #UFC803024) using the FASP method. To the resulting peptide solution, 1.5 mL acetonitrile, 7.5 mL of Incubation Buffer (60% acetonitrile, 3% trifluoroacetic acid), and 2 mg equilibrated TiO₂ beads were sequentially added and mixed, followed by incubation for 60 min. TiO₂ beads were washed with 1 mL of 60% acetonitrile, 3% trifluoroacetic acid, 50 mM citric acid three times (20 min per time) and 1 mL of 80% acetonitrile, 0.1% trifluoroacetic acid for 1 min once. Phosphopeptides were eluted with 100 μL of 50% acetonitrile, 14% ammonium hydroxide and then 100 μL of 80% acetonitrile, 5.6% ammonium hydroxide. Peptide solution resulting from the two elution steps was combined and dried down in a SpeedVac. Enriched phosphopeptides were analyzed using an EASY-nLC 1200 connected to an Orbitrap Fusion Lumos operated in a data-dependent manner. The acquired MS data (9 RAW files) were searched against the Uniprot_Human database (released on 01/22/2016) using MaxQuant (v1.5.5.1). The data have been deposited to the PRIDE (identifier PXD018871). Perseus (v1.6.6.0) was applied to perform quality assessment and statistical analysis, as described in the interactome analysis. Phosphosites meeting the criteria of FDR ≤ 1% and localization probability of >0.75 were accepted as confident identifications. To compute relative phosphorylation level changes, relative phosphosite intensities were normalized against the relative abundance of corresponding proteins, which were quantified in the above-mentioned label-free proteomic analysis. To quantify kinase activity changes in each comparison, a properly formatted dataset containing phosphosites with $p < 0.05$ was inputted into

the KSEA App[52], where the PhosphoSitePlus plus NetworKIN dataset was selected. Only kinases with ≥5 substrate sites were selected for the quantification of kinase activities. For each kinase, the log2(Ctrl/KO) ratios of all substate sites were averaged to infer kinase activity changes.

**Western blot.** Membranes were probed with antibodies against RIPK2 (1:1000, Cell Signaling Technology (CST) #4142 or Santa Cruz Biotechnology #sc-166765), phospho-NF-κB p65 (Ser536) (1;1000, CST #3033), NF-κB p65 (1:1000, CST #8242), IκBα (1:1000, CST #4814), c-Myc (1:5000, Abcam #ab32072), phospho-c-Myc (S62) (1:1000, CST #13748S), ubiquitin (K48-linkage specific) (1:1000, CST #12805), FLAG (1:5000, Sigma #F1804), MKK7 (1:1000, CST #4172S or 1:2000, Santa Cruz #sc-25288), MKK7 (phospho-Ser271) (1:1000, Aviva Systems Biology #OAAF05547), JNK (1:1000, CST #9252S or 1:2000 Santa Cruz sc-7345), JNK (phospho-T183/Y185) (1:1000, CST #9251S), mCherry (1:1000, Abcam #ab213511), GFP (1:1000, Abcam #ab290), β-actin (1:5000, Sigma Aldrich #5441), or GAPDH (1:1000, CST #3683). Signal was visualized with secondary HRP-conjugated antibodies (1:5000, CST #7074S or #7076S) and chemiluminescent detection.

**c-Myc ubiquitination assay.** Control and RIPK2-KO PC3 cells were treated with 10 μM MG132 (Sigma Aldrich #474790) for 4 h before cell lysis. Immunoprecipitation was performed as described in the Interactome analysis section, except that an anti-c-Myc antibody (Abcam #ab32072) was used. Eluted proteins were probed by immunoblotting as indicated.

**Immunofluorescence and fluorescence imaging.** For immunofluorescence imaging of NES-RIPK2m4-3×FLAG or NLS-RIPK2m4-3×FLAG, 2 mL (1.25 × 10⁵/mL) of RIPK2-KO HEK293T cells were grown on poly-L-lysine pre-treated coverslips. Cells were transfected with a plasmid (1.0 μg per plasmid) for 48 h and then fixed, permeabilized, washed, and blocked. The primary antibody against FLAG (1:500, Sigma Aldrich #F1804) was diluted in 2% BSA and incubated at 4 °C overnight. Fluorochrome-conjugated secondary antibody (1:1000, Cell Signaling Technology #4408) was diluted in 2% BSA and incubated for 1 h at 37 °C in the dark. The coverslips were transferred to slides mounted in Mounting Medium with DAPI (Millipore #DUO82040), and the cells were viewed under an All-in-one fluorescence microscope (BZ-X700, Keyence). For fluorescence colocalization imaging of RIPK2 and MKK7, RIPK2/MKK7-DKO HEK293T cells were co-transfected with (1) mCherry and eGFP, (2) MKK7-mCherry and eGFP, (3) mCherry and RIPK2-eGFP, or (4) MKK7-mCherry and RIPK2-eGFP (0.5 μg per plasmid) for 48 h. Cells were fixed and visualized as described in the immunofluorescence imaging. For immunofluorescence colocalization imaging of MKK7 and c-Myc, RIPK2/MKK7-DKO HEK293T or 22Rv1 cells were transfected with RIPK2m4, MYC, and MKK7-His plasmids (0.5 μg per plasmid) for 48 h, fixed, permeabilized, washed, and blocked as described above. The primary antibody against MKK7 (1:200, Santa Cruz #sc-25288) or c-Myc (1:200, Abcam #ab32072) was diluted in 2% BSA and incubated at 4 °C overnight. Cells were incubated with fluorochrome-conjugated secondary antibodies and visualized as described above.

**Proximity ligation assay (PLA).** PLA was performed according to the manufacturer's instruction (Sigma Aldrich, #DUO92101). Primary antibodies were diluted in Duolink antibody diluent as follows: rabbit anti-RIPK2 (Cell Signaling Technology #4142) at 1:500, mouse anti-MKK7 (Santa Cruz Biotechnology #sc-25288) at 1:100, and mouse anti-PRKDC (Santa Cruz Biotechnology #sc-5282) at 1:100. Imaging was performed with an All-in-one fluorescence microscope (BZ-X700, Keyence) under TexasRed and DAPI filters. The numbers of PLA signals (shown in red) and cells (nuclei, shown in blue) from four to five random fields were quantified for each sample by Image J (v1.52p)[72]. For quantitative comparison, the numbers under control conditions are the sum of dots per cell detected using each of the two target-specific antibodies.

**In vitro kinase assay.** For the analysis of RIPK2 phosphorylation of MKK7, kinase-dead (K149A) MKK7 was cloned into a TAT-HA vector (a gift from Steven Dowdy at UCSD). The 6 × His-tagged recombinant protein was expressed in E. coli (BL21-DE3), purified using Ni-NTA affinity chromatography, and eluted using imidazole-based competition. 3 × Flag-tagged RIPK2 was expressed in HeLa cells and immunoprecipitated on Protein A Agarose beads using an anti-Flag antibody. Kinase reaction was performed for 30 min using 0.5 μCi of [γ-³²P]ATP per reaction, and reaction mixtures were subjected to SDS-PAGE followed by exposure for autoradiography according to our published procedure[73]. For the analysis of MKK7 phosphorylation of c-Myc, in a 20 μL reaction system, 250 ng 6×His-c-Myc (Ray-Biotech #230-00580-50) and 500 ng GST-MKK7 (Abnova #P5678) were incubated in 1× kinase buffer (Cell Signaling Technology #9802) supplemented with 500 μM ATP (Cell Signaling Technology #9804) for 30 min at 30 °C. For negative control, MKK7-GST was not added. Proteins were resolved by SDS-PAGE and analyzed by immunoblotting. In another experiment, control vector, MKK7-3E, or MKK7-149A was transiently transfected into RIPK2/MKK7-DKO HEK293T cells. Following cell lysis, MKK7 complexes were enriched by IP using an anti-MKK7 antibody. Beads were washed by IP washing buffer containing 1% Tween-20 for three times and by 1× kinase reaction buffer two times. Subsequently, IP products were incubated with

recombinant c-Myc in kinase reaction buffer for 30 min at room temperature. Proteins were resolved by SDS-PAGE and analyzed by immunoblotting.

**Structural modeling**. The structural model of the RIPK2-MKK7 complex was generated from available structures in the RCSB Protein Data Bank (PDB). Briefly, the 3D structure of the RIPK2 kinase domain was used from an available structure (PDB code 6ES0), and that of the MKK7 kinase domain was generated by homology modeling[74], using the structure of MEK1 (PDB code 1S9I) as a template. Putative dimerization models of kinase domains were generated with Rosetta (v3.11) and ZDOCK (v2.3.2). The top 10 models from each program were further subjected to molecular minimization and 1.2 ns short molecular dynamics using Desmond (Schrodinger, SBGrid Consortium), in order to optimize molecular interaction and estimate energetics of the complex. Finally, the energetically stable protein complex was selected.

**Correlation analysis**. To perform correlation analysis, we analyzed the PCTA prostate cancer ($n = 2115$)[24], TCGA Firehose Legacy prostate cancer ($n = 499$), and TCGA PanCancer Atlas[75] (32 studies) cohort data. For gene expression abundance, PCTA provides median-centered and quantile normalized expression values. For the TCGA Firehose Legacy prostate cancer cohort, median-centered $\log_2(\text{FPKM} + 1)$ values were computed for each gene. For the TCGA PanCancer Atlas, RSEM (RNA-Seq by Expectation Maximization) values were log-transformed using $\log_2(\text{RSEM value} + 1)$ and median centered by genes. Given these expression values, gene-set activation scores were computed by using the weighted $Z$ score method. Spearman's method was used to compute correlation coefficients between genes and/or gene sets, using the PCTA portal (thepcta.org) or R (v3.6.2).

**Kaplan–Meier survival analysis**. Kaplan–Meier survival analyses were performed using the TCGA PanCancer Atlas cohort and the Jain cohort (GSE116918)[76]. The log-rank test was performed to compute hazard ratios and statistical significance of survival difference between groups. The graphs were generated using R (v3.6.2).

**Statistical analysis**. Statistical analyses were performed in R (v3.6.2). Unless specified, all statistical tests were two-sided with a significance level of $p < 0.05$. All statistics and reproducibility information are reported in the figure legends. For animal studies, sample sizes were defined on the basis of our past experience to achieve 80% power. For ethical reasons, the minimum number of animals necessary to achieve the scientific objectives was used.

**Reporting summary**. Further information on research design is available in the Nature Research Reporting Summary linked to this article.

## Data availability

The proteomics, interactome, and phosphoproteomics data generated in this study have been deposited in the PRIDE repository under the accession codes PXD018890, PXD018870, and PXD018871, respectively. The cBioPortal data are available for download from cbioportal.org. The PCTA data are available from thepcta.org. The Pharos data are available from pharos.nih.gov. The MSigDB (v7.4) data are available from https://www.gsea-msigdb.org/gsea/msigdb/. The CCLE data are available from depmap.org. The Uniprot_Human data are available from uniprot.org. Source data are provided with this paper.

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

## Acknowledgements

We thank Drs. Jayoung Kim, Gina (Chia-Yi) Chu, Stephen Freedland, Hisashi Tanaka, and their group members, other members of the Freeman and Di Vizio labs, and Dr. Yanping Wang for helpful suggestions. We are grateful to Drs. Mandana Zandian and Jasmine Wang for lab assistance, as well as the Biobank and Translational Research Shared Resource of Cedars-Sinai Cancer. Cartoons in Figs. 8g and 9 were created with BioRender.com. This work was supported by NCI R01 awards 1R01CA218526 and 1R01CA232574, Cedars-Sinai Development of Prostate Cancer Fund, Cedars-Sinai Precision Health Award, and UCLA CTSI Core Voucher Award [W.Y.] and the Department of Defense (DoD)—Early Investigator Research Award W81XWH-18-1-0476 [Y.Y.].

## Author contributions

W.Y., Y.Y. and B.Z. conceived and designed the study. Y.Y. and B.Z. carried out most experiments. C.Q. contributed to the mouse experiments. A.V. contributed to immunoblotting and PCTA analyses. M.K. and K.S. conducted the radioactive in vitro kinase analysis of RIPK2 phosphorylation of MKK7. A.C. and R.M. performed structural modeling. X.Y., B.S.K. and A.G. contributed to the tissue microarray analysis. Y.-J.L. and S.Y. contributed to the PCTA and statistical analyses. L.E., D.D.V., E.P., N.K. and M.R.F. interpreted results and edited the manuscript. W.Y., Y.Y. and B.Z. performed most data analysis, generated figures, and wrote the manuscript. All authors made intellectual contributions and reviewed the manuscript.

## Competing interests

Cedars-Sinai Medical Center has a pending patent application PCT/US2021/024740 (W.Y.) relevant to this study. The remaining authors declare no competing interests.

## Additional information



