## [Peer Review File · Nature Communications]

Receptor-interacting Protein Kinase 2 (RIPK2) Stabilizes c-Myc and is a Therapeutic Target in Prostate Cancer MetastasisREVIEWER COMMENTS

Reviewer #1 (Remarks to the Author):

Summary

In this article, Yan and colleagues take a logical approach to screen for genes potentially implicated in prostate cancer metastasis and progression with a focus on target druggability. They identify RIPK2 as one such target in prostate cancer, and convincingly show that this gene gains copy number in a subset of tumors, is expressed to a higher degree in more aggressive or advanced disease at the RNA and DNA level, and its higher expression is associated with poor prognostic subtypes as well as inferior clinical outcome. The authors then demonstrate that RIPK2 promotes metastatic behavior both in vitro and in vivo but has a very limited impact on cell proliferation in models tested. To identify the mechanism whereby RIPK2 may impact a pro-metastatic phenotype, they use proteomic analysis to suggest a correlation with MYC activity. Functionally, RIPK2 knockout decreases MYC protein expression, which can be rescued. Through a series of protein interaction experiments, the authors suggest the RIPK2 interacts with MKK7, which is experimentally found to be a c-MYC kinase that promotes stability of that protein. Finally, the authors show that chemical inhibition of RIPK2 suppresses metastatic behavior. Interestingly, they show this benefit after early metastatic lesions are found in mice, despite only a minor impact of RIPK2 on cell proliferation.

Overall, this manuscript is an extraordinary amount of data for a single paper, suggests a novel signaling pathway whereby the trajectory of early prostate cancer metastases could be altered, and offers a large body of proteomic data for the research community at large. Congratulations to the authors for assembling this body of work. See comments below for possible additions to round out or clarify the findings.

Comments/Questions:

1) One key experiment is whether RIPK2 is sufficient to promote metastasis? For example, does overexpressing RIPK2 in RWPE-2 cells increase their anchorage independent growth or tumor formation, or does the same experiment in RWPE-1 cells render them able to grow in an anchorage-independent fashion? These experiments would ideally be done in similar contexts as the experiments showing the necessity of RIPK2 for metastatic behavior, but this is a lot of experiments for an already expansive paper. Some cell line experiments may be adequate to illustrate this point.

2) In the first paragraph of the results, I am confused by the numbers of candidate genes nominated by each of the three selection criteria, specifically why there are different numbers in parenthesis versus the 574, 1755, and 2208 genes listed subsequently. The Venn diagram numbers in 1a add up to the latter set of numbers, not the former (for example there appear in 1a to be 574 genes amplified, not 655). Please clarify.

3) Regarding Figure 1f, I believe the way the cell lines are described is a bit incorrect. RWPE-1 cells are immortalized benign prostate epithelial cells, and RWPE-2 cells are the same cells that were transformed with v-Ki-ras and consequently display some hallmarks of cancer. 22Rv1 cells are a subline from CWR22, which was derived from a primary prostate cancer (that patient did have bone mets), whereas LNCaP, DU145 and PC3 were derived from metastatic biopsies. Please revise the figure and associated text to reflect this concisely, though overall I think the figure point of higher expression in models of more advanced prostate cancer remains.

4) Please include citations specifically demonstrating that c-Myc protein expression is necessary or sufficient for the in vitro cell phenotype readouts in Figure 2.

5) Does RIPK2 KO impact MYC mRNA levels?

6) The authors state that "Collectively, RIPK2 stabilize the c-Myc protein by phosphorylating its S62

residue...". This is somewhat untrue, as subsequent data suggests it does not phosphorylate c-Myc directly, so a slight wording change would be appropriate.

7) Does the kinase dead mutant of MKK7 from figure 6f decrease p-c-MYC-S62?

8) In the discussion, the authors state that the canonical NOD/RIPK2 pathway is probably not important in PC development and progression. This is based exclusively off cell line data and cannot speak to the tumor microenvironment in patients. The data strongly shows a relationship between RIPK2 and pro-metastatic behavior in model systems as well as an association with more aggressive clinical phenotypes, but importance of the canonical pathways in patients cannot be ruled out.

9) In the discussion the authors state that ponatinib is well tolerated. This is a subjective statement, and ponatinib is frequently discontinued in CML therapy due to intolerance, especially due to cardiac issues or fluid retention, which are issues faced by advanced prostate cancer patients. Additionally it is possible that the dose of ponatinib required to impact prostate cancer metastases is higher than that required to impact circulating CML cells. I find RIPK2 to be an interesting target, but am not so sure it would be easily targeted in clinic with ponatinib, especially since the goal is to ameliorate metastatic burden (meaning, it would be given earlier in the course of treatment, not at end-stage).

There are a few spots in the early part of the paper that would benefit from a closer revision of English grammar.

Reviewer #2 (Remarks to the Author):

In this paper, the authors produced the following results.

By filtering published databases, the authors found receptor-interacting protein kinase 2 (RIPK2) has the potential to be a drug target for suppressing prostate cancer metastasis.

The relationship between RIPK2 and PC metastasis is validated by the comparison of PC metastasis and growth between RIPK2-KO cells and the control group.

Gene set enrichment analysis (GSEA) shows that RIPK2 potentially activates c-Myc, which was further validated by their high correlation in the data from PCTA and TCGA cohorts.

The upregulation is achieved by promoting the S62 phosphorylation and increasing the stability of c-Myc.

Interactome analysis and phosphoproteomic comparison showed that RIPK2 potentially activates c-Myc-S62 kinase via intermediate proteins, in which MKK7 pathway was claimed to be the most likely one.

This conclusion was further confirmed by PLA, fluorescence colocalization, co-IP analyses, and MKK7-KO experiments.

Pharmacological inhibition of RIPK2 inactivated RIPK2/MKK7(/JNK)/c-Myc signaling and suppresses PC metastatic outgrowth, proving that RIPK2 is a promising drug target for inhibiting PC metastasis.

Overall the paper is well-written. I do have one question.

In "c-Jun N-terminal kinases (JNKs) downstream of MAP2K7 (also known as MKK7) appeared to be most activated by RIPK2", why does "most activated" indicate it is the correct signal pathway? How likely that the other proteins also correspond to reasonable pathways?

Reviewer #3 (Remarks to the Author):

Yan, Zhou and colleagues present a study on the RIPK2 kinase in prostate cancer metastasis and propose it as a therapeutic target and explain its mechanism in MYC activation by another kinase MKK7. The study includes integration of functional, multi-omics and data analysis components to

understand this potential therapeutic avenue and appears as a strong advance in the field. Some comments and suggestions are listed below.

1. With 8 figures and 39 supplementary figures with a total of ~180 panels, this study appears very dense and some streamlining would help condense the main message.
2. Only cursory statistics are provided throughout. In the initial rationale of selecting RIPK2 as the focus, statistical tests are needed to convey significance: for example, in the second paragraph of results (p4), none of results described in the main figures appear to have associated P-values and effect sizes reported in the text (Fig 1 bcdefg). Significant correlation with Gleason? enrichment of CNAs or mRNA overexpression in metastatic tumors? etc Similar issues could be fixed elsewhere as well, such as the cell line and mouse data in the KO section.
3. Selection of RIPK2 using the three criteria appears ad hoc. Since this is performed across a series of different datasets in Cbio portal one may wonder if the detected trends also hold in most of the individual datasets or only when aggregated? batch effects of these various datasets could affect the reasoning.
4. Section titled "RIPK2 is necessary for PC metastasis" seems overstated because in practice they present some data to support metastatic phenotypes in KO experiments of RIPK2, while necessity itself is not proven.
5. Since MYC and RIPK2 are often co-amplified, it would be important to analyze what other genes in those copy-altered genomic segments respond transcriptionally to CNAs together with RIPK2 & MYC and potentially participate in the RIPK2/MYC/MKK7 pathway. Are any apparent in the other omics analyses they already perform?
6. The use of HEK293 cell line for RIPK2 & MKK7 double KO and immunofluorescence and proteomics experiments is not well rationalized since this is an embryonic kidney cell line. a prostate cancer cell line would have been more appropriate.

minor

A. page and line numbers would help reviewers

B. "proteomic analysis identified 5,237 protein groups" - do the author mean proteins not protein groups?

Reviewer #4 (Remarks to the Author):

The authors of the current manuscript have employed a multiomics bioinformatic data mining of public repositories to find potentially actionable targets in the context of prostate cancer metastasis.

Important that initial bioinformatic step may be, the main value of the paper in my view lies in the investigative experimental follow up undertaken to find and elucidate the non-canonical RIPK2/MKK7/MYC pathway, and showing how important that pathway is to metastasis in prostate cancer. Moreover, analysis of public data suggests that this new pathway is relevant to MYC activity in a pan cancer fashion.

The use of in vitro cell assays, animal studies, three varieties of proteomics (shotgun label free, immunoprecipitation and phospho), among other methods, is put to good use in elucidating the RIPK2/MKK7/MYC pathway in a comprehensive manner. A bit more detail is needed to explain some of the methods however.

The statistical analyses are appropriate. The results section is written in a structured manner, and abstract, introduction and conclusions seem appropriate.

I have a few minor comments:

1. I think the use of the many different publicly available datasets warrants its own part in the Methods section. It would be good to describe, for each resource, the experimental data type (e.g. transcriptomics by RNAseq, genomic mutation data, or druggable genome, etc), cohort name and size, any reference publications, URLs, or any other important information. I should say I got confused initially with the first appearance of the TCGA cohort (results, page 6) in the main text, as it is mentioned without any reference, and shows up in parallel to the PCTA cohort, which itself had a proper introduction (including a publication reference) at beginning of the Results section on page 4. From the figure legends it became clear that this is the Broad Institute Firehose Legacy cohort (as I am unfamiliar with these cohorts, it is still unclear to me if this cohort relates to or is separate from the TCGA PanCancer Atlas). Such information should be stated in the main text, or therein a referral should be made to a neatly organized “public resources” Methods section, as suggested above.
2. Results, page 4, where it reads “we applied three stringent criteria to filter large-scale clinical omics databases” it should instead read “we applied stringent criteria to filter three large-scale clinical omics databases”. Otherwise, it sounds like the criteria are the clinical omics databases.
3. Results, page 5: “A total of 574, 1,755, and 2,208 human genes meet the three criteria”. I understand the intended meaning, but rephrasing is needed, as indeed only seven genes actually meet the three criteria.
4. Results, page 7: “Together, RIPK2 is required...” should instead be: “Taken together, these results indicate that RIPK2 is required...” or something to that effect. Similar issue is encountered also in Results, page 11 “Together, RIPK2 positively...”, page 12 “Collectively, RIPK2 stabilizes...”, page 14 “Together, RIPK2 may...”, page 16 “Together, MKK7 is...”, and page 18 “Together, both GSK583...”
5. In the label free proteomics analyses, I can deduce that 6 replicates were used per sample (from the 24 raw files), but it would be good to explicitly state (at least in the Methods section) the number of replicates used, and if these were mere technical injections, or if they were from 6 individual cell cultures. Also, in addition to mentioning the use of FASP in the Methods section, more details are needed, such as approximate number of cells harvested per condition/replicate, approximate amount of protein used for digestion, was trypsin the only enzyme used or some sort of LysC/trypsin combination, what filter units were used, etc.
6. Results, page 7. “...we analyzed three PC3 single-cell clones”. When I first read single cell here, my mind wrongly jumped to single cell proteomics techniques. Of course, that is not what is used here. The single cell clones are surely expanded in culture, and then shotgun proteomics is carried out. But perhaps adding a few words here might help readers not to fall into this mental trap.
7. Results, page 8. “Comprehensive proteomic analysis...” I don’t think “comprehensive” is correct here, given that only about 5000 proteins were identified/quantified. Sure, 5000 is a good analysis depth for single shot proteomics, but I think the word comprehensive should be reserved for prefractionated proteomics, where the number of proteins identified is closer to 10000.
8. In Supp. Fig. 10 one can see the fold change (FC) cut off used to determine up and downregulations in the label free proteomics. It was ± 0.54 (log2 scale). How was that number determined? This FC cut off should be stated, if not in the main text, at least in the methods section.
9. Results, page 9. “...whereas RIPK2 overexpression significantly increased...” How was the RIPK2 overexpression carried out in cells? Was it by use of the RIPK2m4 plasmid, which one encounters downstream in the main text? Please clarify.
10. Results, page 10. “...RIPK2 and MYC signature genes...” Please define the gene signatures in the main text, not only in the legend of Fig.3g.
11. Results, page 12. Please remove or substitute “Nevertheless”.
12. Results, page 18. “...followed by daily treatment with vehicle control or GSK583.” Please state here the dosage applied (10 mg/kg/day, as stated in legend of Fig.8h).
13. In Methods, please correct PXD accession numbers. All are missing a 0 after PXD. Namely, PXD18890, PXD18870 and PXD18871 should be corrected to PXD018890, PXD018870 and PXD018871.

Rui MM Branca

REVIEWER COMMENTS

Reviewer #1 (Remarks to the Author):

Summary

In this article, Yan and colleagues take a logical approach to screen for genes potentially implicated in prostate cancer metastasis and progression with a focus on target druggability. They identify RIPK2 as one such target in prostate cancer, and convincingly show that this gene gains copy number in a subset of tumors, is expressed to a higher degree in more aggressive or advanced disease at the RNA and DNA level, and its higher expression is associated with poor prognostic subtypes as well as inferior clinical outcome. The authors then demonstrate that RIPK2 promotes metastatic behavior both in vitro and in vivo but has a very limited impact on cell proliferation in models tested. To identify the mechanism whereby RIPK2 may impact a pro-metastatic phenotype, they use proteomic analysis to suggest a correlation with MYC activity. Functionally, RIPK2 knockout decreases MYC protein expression, which can be rescued. Through a series of protein interaction experiments, the authors suggest the RIPK2 interacts with MKK7, which is experimentally found to be a c-MYC kinase that promotes stability of that protein. Finally, the authors show that chemical inhibition of RIPK2 suppresses metastatic behavior. Interestingly, they show this benefit after early metastatic lesions are found in mice, despite only a minor impact of RIPK2 on cell proliferation.

Overall, this manuscript is an extraordinary amount of data for a single paper, suggests a novel signaling pathway whereby the trajectory of early prostate cancer metastases could be altered, and offers a large body of proteomic data for the research community at large. Congratulations to the authors for assembling this body of work. See comments below for possible additions to round out or clarify the findings.

We thank the reviewer for the positive comments!

Comments/Questions:

1) One key experiment is whether RIPK2 is sufficient to promote metastasis? For example, does overexpressing RIPK2 in RWPE-2 cells increase their anchorage independent growth or tumor formation, or does the same experiment in RWPE-1 cells render them able to grow in an anchorage-independent fashion? These experiments would ideally be done in similar contexts as the experiments showing the necessity of RIPK2 for metastatic behavior, but this is a lot of experiments for an already expansive paper. Some cell line experiments may be adequate to illustrate this point.

Thanks for the suggestions! We stably overexpressed RIPK2 in both tumorigenic RWPE-2 and immortalized but not tumorigenic RWPE-1 cells via lentiviral transfection. Indeed, enforced expression of RIPK2 in RWPE-2 cells significantly increased their anchorage-independent growth (new Fig. 2f-2g; also see Fig. R1 below), supporting that RIPK2 promotes metastasis.

Fig. R1. RIPK2 promotes the anchorage-independent colony formation of RWPE-2 cells. **a** Representative immunoblots of the indicated proteins in control and stable *RIPK2*-overexpressing (*RIPK2*-OE) RWPE-2 cells. **b** Representative images of soft agar assays of control and *RIPK2*-OE RWPE-2 cells. **c** Quantification of soft agar assay results.

In comparison, neither control nor *RIPK2*-overexpressing RWPE-1 cells could grow in an anchorage-independent fashion (Fig. R2). The finding suggests that *RIPK2* alone may be insufficient to transform RWPE-1. This is consistent with previous studies showing that early passage RWPE-1 cells overexpressing Pim1 (a direct c-Myc kinase¹) could not form colonies in soft agar assay or tumors in nu/nu nude mice.²

Fig. R2. Neither control nor *RIPK2*-overexpressing RWPE-1 cells could grow in an anchorage-independent fashion. **a** Representative immunoblots of the indicated proteins in control and stable *RIPK2*-OE RWPE-1 cells. **b** Representative images of soft agar assays of control and *RIPK2*-OE RWPE-1 cells. Cells were cultured on soft agar for three weeks in three biological replicates.

References

1. Zhang, Y., Wang, Z., Li, X. & Magnuson, N. S. Pim kinase-dependent inhibition of c-Myc degradation. *Oncogene* 27, 4809–4819 (2008).
2. Kim, J., Roh, M. & Abdulkadir, S. A. Pim1 promotes human prostate cancer cell tumorigenicity and c-MYC transcriptional activity. *BMC Cancer* 10, 248 (2010).

2) In the first paragraph of the results, I am confused by the numbers of candidate genes nominated by each of the three selection criteria, specifically why there are different numbers in parenthesis versus the 574, 1755, and 2208 genes listed subsequently. The Venn diagram numbers in 1a add up to the latter set of numbers, not the former (for example there appear in 1a to be 574 genes amplified, not 655). Please clarify.

We apologize for the confusion. The numbers in parentheses indicate the numbers of prostate cancer specimens and not of genes. To avoid confusion, we replaced “n=655” with “655 specimens” and “n=2,115” with “2,115 specimens” (new Fig. 1a, also see Fig. R3 below). We also added “Σ=655”, “Σ=1,655” (1,755 in the original manuscript was a typo), and “Σ=2,208” to show the total numbers of genes meeting the three criteria, respectively. Source data for Fig. 1a were provided in the resubmission.

Fig. R3. Venn diagram of human genes meeting the indicated criteria.

3) Regarding Figure 1f, I believe the way the cell lines are described is a bit incorrect. RWPE-1 cells are immortalized benign prostate epithelial cells, and RWPE-2 cells are the same cells that were transformed with v-Ki-ras and consequently display some hallmarks of cancer. 22Rv1 cells are a subline from CWR22, which was derived from a primary prostate cancer (that patient did have bone mets), whereas LNCaP, DU145 and PC3 were derived from metastatic biopsies. Please revise the figure and associated text to reflect this concisely, though overall I think the figure point of higher expression in models of more advanced prostate cancer remains.

We apologize for the inaccurate description. We were aware that the 22Rv1 cell line was originally derived from a primary prostate cancer and meant to categorize the cell lines based on whether they are capable of developing metastasis in mice. Many studies, including ours, showed that 22Rv1 cells can form metastases in immunocompromised mice.¹⁻⁴ In comparison, RWPE-1 and RWPE-2 do not form metastasis in mice.⁴ The revised text (lines 109-113) now reads:

“Consistent with the clinical findings, RIPK2 protein abundance is substantially higher in PC cell line models that are capable of forming metastases in immunocompromised mice, such as PC3, DU145, 22Rv1, and LNCaP, compared with non-metastatic prostate cell lines such as RWPE-2 and RWPE-1 (Fig. 1f).”

References

1. Rotinen, M. et al. ONECUT2 is a targetable master regulator of lethal prostate cancer that suppresses the androgen axis. *Nat. Med.* 24, 1887–1898 (2018).
2. Tsai, C. H. et al. Metastatic progression of prostate cancer is mediated by autonomous binding of galectin-4-O-glycan to cancer cells. *Cancer Res.* 76, 5756–5767 (2016).
3. Drake, J. M., Gabriel, C. L. & Henry, M. D. Assessing tumor growth and distribution in a model of prostate cancer metastasis using bioluminescence imaging. *Clin. Exp. Metastasis* 22, 674–684 (2005).
4. Cunningham, D. & You, Z. In vitro and in vivo model systems used in prostate cancer research. *J. Biol. Methods* 2, 17 (2015).

4) Please include citations specifically demonstrating that c-Myc protein expression is necessary or sufficient for the in vitro cell phenotype readouts in Figure 2.

Representative references for each phenotype are listed as follows:

A. c-Myc in cell invasion

1. Ellwood-Yen, K. et al. Myc-driven murine prostate cancer shares molecular features with human prostate tumors. *Cancer Cell* 4, 223–238 (2003).
2. Kim, J. et al. A mouse model of heterogeneous, c-MYC-initiated prostate cancer with loss of Pten and p53. *Oncogene* 31, 322–332 (2012).
3. Benassi, B. et al. MYC is activated by USP2a-mediated modulation of MicroRNAs in prostate cancer. *Cancer Discov.* 2, 236–247 (2012).

B. c-Myc in anchorage-dependent colony formation

1. Bernard, D. & Pourtier-Manzanedo, a. Myc confers androgen-independent prostate cancer cell growth. *J. Clin. Invest.* 112, 1724–1731 (2003).
2. Napoli, S., Pastori, C., Magistri, M., Carbone, G. M. & Catapano, C. V. Promoter-specific transcriptional interference and c-myc gene silencing by siRNAs in human cells. *EMBO J.* 28, 1708–1719 (2009).
3. Ciccarelli, C. et al. Disruption of MEK/ERK/c-Myc signaling radiosensitizes prostate cancer cells in vitro and in vivo. *J. Cancer Res. Clin. Oncol.* 144, 1685–1699 (2018).

C. c-Myc in anchorage-independent colony formation

1. Bernard, D. & Pourtier-Manzanedo, a. Myc confers androgen-independent prostate cancer cell growth. *J. Clin. Invest.* 112, 1724–1731 (2003).
2. Fan, L. et al. Regulation of c-Myc expression by the histone demethylase JMJD1A is essential for prostate cancer cell growth and survival. *Oncogene* 35, 2441–2452 (2016).
3. Zhang, Y., Wang, Z., Li, X. & Magnuson, N. S. Pim kinase-dependent inhibition of c-Myc degradation. *Oncogene* 27, 4809–4819 (2008).
4. Kalkat, M. et al. MYC protein interactome profiling reveals functionally distinct regions that cooperate to drive tumorigenesis. *Mol. Cell* 72, 836-848.e7 (2018).
5. Kim, J., Roh, M. & Abdulkadir, S. A. Pim1 promotes human prostate cancer cell tumorigenicity and c-MYC transcriptional activity. *BMC Cancer* 10, 248 (2010).

We incorporated the references into the “RIPK2 upregulates the c-Myc protein by phosphorylating c-Myc-S62 and preventing c-Myc from proteasomal degradation” section. The text (lines 229-232) now reads:

“*Various studies have shown that the c-Myc oncoprotein is necessary or sufficient for cancer cell invasion*³⁶⁻³⁸, *anchorage-dependent colony formation*³⁸⁻⁴⁰, *anchorage-independent colony formation*⁴¹⁻⁴⁵, *and metastasis*¹²⁻²⁰.”

5) Does RIPK2 KO impact MYC mRNA levels?

To answer the question, we performed qPCR analyses of control and *RIPK2*-KO PC3 and 22Rv1 cells. Although *RIPK2*-KO decreased *MYC* mRNA levels (by ~47%) in PC3 cells, it did not significantly change *MYC* mRNA levels in 22Rv1 cells (new Supplementary Fig. 18a; also see Fig. R4 below). In addition, ectopic overexpression of *RIPK2m4* in *RIPK2*-KO PC3 cells only marginally increased *MYC* mRNA levels. Therefore, we conclude that RIPK2 generally has a

negligible effect on regulating *MYC* mRNA levels. The reduction of *MYC* mRNA levels in *RIPK2*-KO PC3 cells is likely a secondary event.

Fig. R4. The regulation of c-Myc by RIPK2 is mainly not at the transcriptional level. Bar plot of *MYC* mRNA level changes (normalized to GAPDH) caused by transient overexpression of *RIPK2m4* in *RIPK2*-KO PC3 cells, by *RIPK2*-KO in PC3 cells, or by *RIPK2*-KO in 22Rv1 cells. Data are Mean ± SD; unpaired two-tailed Student's *t*-test.

We incorporated the information into the revised manuscript. The revised text (lines 253-260) now reads:

“Using quantitative real-time polymerase chain reaction (RT-PCR), we found that transient *RIPK2* overexpression in PC3 cells, *RIPK2*-KO in 22Rv1 cells, or *RIPK2* inhibition by GSK583 (a *RIPK2*-selective inhibitor) at different doses and time only modestly (~10%) regulated *MYC* mRNA abundance (Supplementary Fig. 18). In comparison, *RIPK2*-KO in PC3 cells caused a strong reduction in *MYC* mRNA levels (by ~47%) (Supplementary Fig. 18a). However, this is likely a secondary event, considering that transient *RIPK2*-OE or *RIPK2* inhibition in PC3 cells only marginally modulated *MYC* mRNA levels (Supplementary Fig. 18). Collectively, the regulation of c-Myc protein abundance by *RIPK2* is mainly via a post-transcriptional mechanism.”

6) The authors state that “Collectively, *RIPK2* stabilize the c-Myc protein by phosphorylating its S62 residue...”. This is somewhat untrue, as subsequent data suggests it does not phosphorylate c-Myc directly, so a slight wording change would be appropriate.

Thanks for pointing this out! The revised text (lines 275-276) now reads:

“... *RIPK2* stabilizes the c-Myc protein by either directly or indirectly phosphorylating its S62 residue ...”

7) Does the kinase dead mutant of MKK7 from figure 6f decrease p-c-MYC-S62?

To answer the question, we transfected control vector, constitutively active MKK7-3E (*i.e.*, MKK7-S271E, T275E, S277E), or kinase-dead MKK7-K149A into *RIPK2*/MKK7-double-knockout HEK293T cells, followed by immunoprecipitation (IP) using an anti-MKK7 antibody. We then performed *in vitro* kinase assays by incubating the IP products with a recombinant c-Myc protein. Our result showed that MKK7-K149A is much less efficient than MKK7-3E in phosphorylating c-Myc-S62 (new Supplementary Fig. S33; also see Fig. R5 below). The residual c-Myc-S62 phosphorylation under the K149A condition is likely due to the co-immunoprecipitation of other direct c-Myc-S62 kinases or due to the residual kinase activity of MKK7-K149A.

Fig. R5. Kinase-dead MKK7-K149A is much less efficient than constitutively active MKK7-3E in phosphorylating c-Myc-S62. **a** Representative immunoblots of the indicated proteins after *in vitro* kinase assay reactions. **b** Quantification of relative c-Myc-S62 phosphorylation levels under the indicated conditions.

The revised text (lines 375-380) now reads:

“In vitro kinase assays showed that recombinant MKK7 could directly phosphorylate c-Myc-S62 (Fig. 7d) and that immunoprecipitated constitutively active MKK7-3E (S271E, T275E, S277E) is much more efficient than immunoprecipitated kinase-dead MKK7-K149A in phosphorylating c-Myc-S62 (Supplementary Fig. S33). The residual c-Myc-S62 phosphorylation under the K149A condition is likely due to the co-immunoprecipitation of other direct c-Myc-S62 kinases or the residual kinase activity of MKK7-K149A.”

8) In the discussion, the authors state that the canonical NOD/RIPK2 pathway is probably not important in PC development and progression. This is based exclusively off cell line data and cannot speak to the tumor microenvironment in patients. The data strongly shows a relationship between RIPK2 and pro-metastatic behavior in model systems as well as an association with more aggressive clinical phenotypes, but importance of the canonical pathways in patients cannot be ruled out.

We agree with the reviewer. The revised text (lines 493-494) now reads:

“Nevertheless, these are based on in vitro cell culture models, so the importance of the canonical NOD/RIPK2 pathway in PC patients cannot yet be ruled out.”

9) In the discussion the authors state that ponatinib is well tolerated. This is a subjective statement, and ponatinib is frequently discontinued in CML therapy due to intolerance, especially due to cardiac issues or fluid retention, which are issues faced by advanced prostate cancer patients. Additionally it is possible that the dose of ponatinib required to impact prostate cancer metastases is higher than that required to impact circulating CML cells. I find RIPK2 to be an interesting target, but am not so sure it would be easily targeted in clinic with ponatinib, especially since the goal is to ameliorate metastatic burden (meaning, it would be given earlier in the course of treatment, not at end-stage).

While we appreciate the concerns about the toxicity of ponatinib, the consensus in the community of hematologic malignancies is that ponatinib is an agent with manageable adverse events¹⁻³. Studies showed that the side effects of ponatinib are generally dose dependent.¹ The phase 2 PACE trial (NCT01207440) showed that although 45 mg daily ponatinib caused treatment-emergent arterial occlusive events (AOEs) in 42% of patients, 30 mg or 15 mg daily ponatinib caused much less frequent (24% and 26%, respectively) AOEs.² The study concluded that “tolerability was acceptable in this heavily pretreated population with 5 years of follow-up.”

We agree that the dose of ponatinib required to impact prostate cancer metastasis might be higher than that in CML treatment. Nonetheless, ponatinib has a known toxicity profile and management

strategy can be adopted from the CML literature and experience. To maximize the benefit-risk profile of ponatinib in CP-CML, a strategy applied in the OPTIC trial (NCT02467270) is to initiate treatment at 45 mg daily and then reduce the dose to 15 mg daily after $\leq 1\%$ BCR-ABL1 is achieved.³ The dosing reduction strategy may be adopted to improve the benefit-risk profile of ponatinib in non-metastatic castration-resistant prostate cancer (nmCRPC).

To avoid any overstatement, we modified the sentence. The text (lines 473-480) now reads:
“These results also have high translational value because RIPK2 can be inhibited by ponatinib (Iclusig) – an FDA-approved agent whose safety profiles are known. Of note, when being administered at 30 mg orally once daily, ponatinib is tolerated and suitable for such a metastasis-inhibition strategy. The benefit-risk profile of ponatinib might be further improved by initiating treatment at 45 mg daily and then reducing the dose to 15 mg daily, a strategy applied in the OPTIC trial (NCT02467270).”

References

1. Chan, O. et al. Side-effects profile and outcomes of ponatinib in the treatment of chronic myeloid leukemia. *Blood Adv.* 4, 530–538 (2020).
2. Cortes, J. E. et al. Ponatinib efficacy and safety in Philadelphia chromosome-positive leukemia: final 5-year results of the phase 2 PACE trial. *Blood* 132, 393–404 (2018).
3. Kantarjian, H. M. et al. Efficacy and safety of ponatinib (PON) in patients with chronic-phase chronic myeloid leukemia (CP-CML) who failed one or more second-generation (2G) tyrosine kinase inhibitors (TKIs): Analyses based on PACE and OPTIC. *Blood* 136, 43–44 (2020).

There are a few spots in the early part of the paper that would benefit from a closer revision of English grammar.

Thanks for pointing this out! We have carefully proofread the revised manuscript to minimize grammatical errors and typos.

Reviewer #2 (Remarks to the Author):

In this paper, the authors produced the following results.

By filtering published databases, the authors found receptor-interacting protein kinase 2 (RIPK2) has the potential to be a drug target for suppressing prostate cancer metastasis.

The relationship between RIPK2 and PC metastasis is validated by the comparison of PC metastasis and growth between RIPK2-KO cells and the control group.

Gene set enrichment analysis (GSEA) shows that RIPK2 potentially activates c-Myc, which was further validated by their high correlation in the data from PCTA and TCGA cohorts.

The upregulation is achieved by promoting the S62 phosphorylation and increasing the stability of c-Myc.

Interactome analysis and phosphoproteomic comparison showed that RIPK2 potentially activates c-Myc-S62 kinase via intermediate proteins, in which MKK7 pathway was claimed to be the most likely one.

This conclusion was further confirmed by PLA, fluorescence colocalization, co-IP analyses, and MKK7-KO experiments.

Pharmacological inhibition of RIPK2 inactivated RIPK2/MKK7(/JNK)/c-Myc signaling and suppresses PC metastatic outgrowth, proving that RIPK2 is a promising drug target for inhibiting PC metastasis.

Overall the paper is well-written. I do have one question.

We thank the reviewer for the positive feedback!

In “c-Jun N-terminal kinases (JNKs) downstream of MAP2K7 (also known as MKK7) appeared to be most activated by RIPK2”, why does “most activated” indicate it is the correct signal pathway? How likely that the other proteins also correspond to reasonable pathways?

The “most activated” pathway was identified by an integrated analysis of our interactome and phosphoproteomics datasets. The “most activated” pathway does not necessarily indicate the only correct signaling pathway; however, it does represent a strong candidate for validation. In our original manuscript, we concluded that *“Together, the findings suggest that RIPK2 may indirectly phosphorylate c-Myc-S62 via multiple kinase pathways, particularly the MKK7 pathway.”*

Importantly, by performing a series of experiments, we confirmed that MKK7 is the major mediator of RIPK2’s indirect phosphorylation of c-Myc (see the “MKK7 is a major mediator of RIPK2 regulation of c-Myc” section in the manuscript). We also noted that MKK7 is not the only mediator; the other proteins in the signaling network may be additional (but minor) mediators of RIPK2’s indirect phosphorylation of c-Myc. In our manuscript (now lines 353-354), we stated that *“...MKK7 is a major (albeit not the only) mediator of RIPK2 regulation of c-Myc.”*

Reviewer #3 (Remarks to the Author):

Yan, Zhou and colleagues present a study on the RIPK2 kinase in prostate cancer metastasis and propose it as a therapeutic target and explain its mechanism in MYC activation by another kinase MKK7. The study includes integration of functional, multi-omics and data analysis components to understand this potential therapeutic avenue and appears as a strong advance in the field. Some comments and suggestions are listed below.

We thank the reviewer for commenting that our study “appears as a strong advance in the field”!

1. With 8 figures and 39 supplementary figures with a total of ~180 panels, this study appears very dense and some streamlining would help condense the main message.

We agree that a large number of figure panels were provided. However, this is because our study is expansive and many experiments were performed to reach solid conclusions. Nonetheless, because we provided Source Data in the resubmission, we deleted the original Supplementary Fig. 5b, 36c, 37b lower panel, and 39c, which correspond to the revised Fig. 2h, Fig. 8h, Supplementary Fig. 38b upper panel, and 8j, respectively. The only difference is that the former were presented as bar plots (including individual data points) whereas the latter were presented as line plots.

2. Only cursory statistics are provided throughout. In the initial rationale of selecting RIPK2 as the focus, statistical tests are needed to convey significance: for example, in the second paragraph of results (p4), none of results described in the main figures appear to have associated P-values and effect sizes reported in the text (Fig 1 bcdefg). Significant correlation with Gleason? enrichment of CNAs or mRNA overexpression in metastatic tumors? etc Similar issues could be fixed elsewhere as well, such as the cell line and mouse data in the KO section.

Thanks for the suggestion! In the revised manuscript, we provided p values and effect sizes for Fig. 1 (also see Source Data). We also provided p values for the other sections.

3. Selection of RIPK2 using the three criteria appears ad hoc. Since this is performed across a series of different datasets in Cbio portal one may wonder if the detected trends also hold in most of the individual datasets or only when aggregated? batch effects of these various datasets could affect the reasoning.

To address the reviewer's concern, we plotted the copy number amplifications of the seven overlapping genes in the three mCRPC cohorts (new Supplementary Fig. 1a, also see Fig. R6 below). The Robinson cohort tends to have the lowest amplification frequencies for the indicated genes. With the exception of PMVK, the detected trends hold in at least two of the three individual datasets. Accordingly, the revised text (lines 88-91) now reads:

“Seven genes meet all three criteria, representing candidate druggable targets of PC metastasis (Fig. 1a). Except for PMVK, these genes are amplified in 10%-26% of PC tissue specimens in at least two out of the three mCRPC cohorts (Supplementary Fig. 1a).”

Fig. R6. Bar plot of the copy number amplification frequencies of the indicated genes in three independent mCRPC cohorts and on average.

4. Section titled "RIPK2 is necessary for PC metastasis" seems overstated because in practice they present some data to support metastatic phenotypes in KO experiments of RIPK2, while necessity itself is not proven.

To address the concern, we replaced “necessary” with “required”. In addition, our new data showed that RIPK2 overexpression increased the anchorage-independent growth of RWPE-2 cells (new Fig. 2f-2g; also see Fig. R1 above).

5. Since MYC and RIPK2 are often co-amplified, it would be important to analyze what other genes in those copy-altered genomic segments respond transcriptionally to CNAs together with RIPK2 & MYC and potentially participate in the RIPK2/MYC/MKK7 pathway. Are any apparent in the other omics analyses they already perform?

This is an interesting and important point. It has been well recognized that chromosome 8q gains are frequent in advanced prostate cancers. Using neXtProt, a comprehensive human protein database, we retrieved a total of 390 genes for which at least one protein product has been credibly identified by mass spectrometry or by a direct biochemical assay. Among these, only proteins encoded by three genes (*i.e.*, ATP6V1H, MCM4, and PRKDC) were identified as RIPK2-interacting proteins. The information was incorporated into new Supplemental Table 3 (column BE). It would be interesting to determine whether the proteins cooperate with RIPK2 in promoting prostate cancer progression and metastasis in the near future. In addition, among proteins encoded by the 390 8q genes, three (DERL1, PTDSS1, and SDCBP) are significantly downregulated, whereas 10 (C8orf82, COMM5, ERICH5, NBN, OPLAH, OXR1, PLEC, STMN2, THEM6, and ZHX2) were significantly upregulated, by *RIPK2*-KO in PC3 cells (new Supplemental Table 2, column CB). Nevertheless, it remains unclear whether the 13 proteins are involved in the RIPK2/MKK7/c-Myc pathway.

6. The use of HEK293 cell line for RIPK2 & MKK7 double KO and immunofluorescence and proteomics experiments is not well rationalized since this is an embryonic kidney cell line. a prostate cancer cell line would have been more appropriate.

HEK293T cells were used for these experiments because they offer substantially higher transfection efficiency than prostate cancer cell lines. In addition, HEK293T cells were used for mechanistic studies and we expect that the results are cell line-independent. To address the concern, we repeated the *RIPK2* & *MKK7* double KO and immunofluorescence experiments in 22Rv1 cells, and the results corroborated the findings in HEK293T cells (new Fig. 6j, 6k, and 7c). We did not attempt to repeat the interactome profiling experiment in prostate cancer cells, because the experiment is complex and high transfection efficiency is required to provide a sufficient amount of input material for comprehensive IP-MS analyses. Nonetheless, in our original manuscript, we validated the association of RIPK2 and MKK7 in three different prostate cancer cell lines (*i.e.*, PC3, DU145, and 22Rv1).

minor

A. page and line numbers would help reviewers

We added page and line numbers to the revised manuscript.

B. "proteomic analysis identified 5,237 protein groups" - do the author mean proteins not protein groups?

These are protein groups. To avoid confusion, we added one sentence to the revised manuscript (lines 179-181):

“Of note, in bottom-up proteomics analysis, different proteins identified by the same set of shared peptides cannot be distinguished, so they are collapsed into a “protein group” to minimize redundant identifications.”

Reviewer #4 (Remarks to the Author):

The authors of the current manuscript have employed a multiomics bioinformatic data mining of public repositories to find potentially actionable targets in the context of prostate cancer metastasis. Important that initial bioinformatic step may be, the main value of the paper in my view lies in the investigative experimental follow up undertaken to find and elucidate the non-canonical RIPK2/MKK7/MYC pathway, and showing how important that pathway is to metastasis in prostate cancer. Moreover, analysis of public data suggests that this new pathway is relevant to MYC activity in a pan cancer fashion.

The use of in vitro cell assays, animal studies, three varieties of proteomics (shotgun label free, immunoprecipitation and phospho), among other methods, is put to good use in elucidating the RIPK2/MKK7/MYC pathway in a comprehensive manner. A bit more detail is needed to explain some of the methods however.

The statistical analyses are appropriate. The results section is written in a structured manner, and abstract, introduction and conclusions seem appropriate.

We thank the reviewer for the positive comments! More details were provided in the revised manuscript according to the reviewer's suggestions (also see below for the responses and changes).

I have a few minor comments:

1. I think the use of the many different publicly available datasets warrants its own part in the Methods section. It would be good to describe, for each resource, the experimental data type (e.g. transcriptomics by RNAseq, genomic mutation data, or druggable genome, etc), cohort name and size, any reference publications, URLs, or any other important information. I should say I got confused initially with the first appearance of the TCGA cohort (results, page 6) in the main text, as it is mentioned without any reference, and shows up in parallel to the PCTA cohort, which itself had a proper introduction (including a publication reference) at beginning of the Results section on page 4. From the figure legends it became clear that this is the Broad Institute Firehose Legacy cohort (as I am unfamiliar with these cohorts, it is still unclear to me if this cohort relates to or is separate from the TCGA PanCancer Atlas). Such information should be stated in the main text, or therein a referral should be made to a neatly organized "public resources" Methods section, as suggested above.

We apologize for the lack of clarity and thanks for the helpful suggestion. Accordingly, we clarified this in the main text and added a "Public Resources" section to the Methods. The new section (lines 513-530) reads as follows.

"Public resources. Publicly accessible web portals, such as the cBioPortal²³, the PCTA²⁴, and the Pharos for human druggable genome²⁵, were used to analyze clinical omics datasets. The cBioPortal (cbioportal.org) hosts omics data from large consortium efforts and publications from individual labs. In the cBioPortal environment, the copy number alterations in three mCRPC cohorts (Grasso et al., n=61; Abida et al., n=444; and Robinson et al., n=150)²⁶⁻²⁸ were analyzed to calculate the alteration frequencies, and the TCGA PanCancer Atlas transcriptomics profiles were analyzed for Kaplan-Meier survival analysis and to compute RIPK2 and MYC activity Z scores. The PCTA portal (thepcta.org) contains the PCTA dataset²⁴, which comprises transcriptomic data from 38 PC cohorts, as well as the TCGA Firehose Legacy dataset⁶³, which contains transcriptomic profiles from 499 PC specimens. The TCGA PanCancer Atlas and Firehose Legacy datasets are quite similar but the former was batch-corrected⁶³. In the PCTA

environment, the “Expression View” function was used to visualize the expression trends of genes or gene lists and the expression levels of RIPK2 in different PCS and PAM50 subtypes, while the “Correlation View” function was used to calculate the correlation coefficients between gene expression levels and MYC activity Z scores and between RIPK2 and MYC activity Z scores. The Pharos (pharos.nih.gov) is a comprehensive and integrated knowledge-base for the Druggable Genome program. It was used to retrieve genes for which clinical-grade (Tclin) and chemical-grade (Tchem, with IC50 < 30 nM) inhibitors have already been developed.”

2. Results, page 4, where it reads “we applied three stringent criteria to filter large-scale clinical omics databases” it should instead read “we applied stringent criteria to filter three large-scale clinical omics databases”. Otherwise, it sounds like the criteria are the clinical omics databases. We modified the text as the reviewer suggested. Now the text (line 80) reads “...we applied stringent criteria to filter three large-scale clinical omics databases...”.

3. Results, page 5: “A total of 574, 1,755, and 2,208 human genes meet the three criteria”. I understand the intended meaning, but rephrasing is needed, as indeed only seven genes actually meet the three criteria.

Sorry for the confusion. In our original manuscript, we wrote that “A total of 574, 1,755, and 2,208 human genes meet the three criteria, respectively, with an overlap of seven genes, whose encoded proteins represent candidate druggable targets of PC metastasis (Fig. 1a).” To minimize confusion, we modified the sentence, which now reads (lines 87-90):

“A total of 574, 1,655, and 2,208 human genes meet the three criteria, respectively (Fig. 1a). Seven genes meet all three criteria, representing candidate druggable targets of PC metastasis (Fig. 1a).”

We also provided the source data for Fig.1a. In the original manuscript, “1,755” was a typo and thus corrected to “1,655” in the revised manuscript.

4. Results, page 7: “Together, RIPK2 is required...” should instead be: “Taken together, these results indicate that RIPK2 is required...” or something to that effect. Similar issue is encountered also in Results, page 11 “Together, RIPK2 positively...”, page 12 “Collectively, RIPK2 stabilizes...”, page 14 “Together, RIPK2 may...”, page 16 “Together, MKK7 is...”, and page 18 “Together, both GSK583...”

We modified the text as the reviewer suggested. Now the texts read as follows.

“Taken together, these results indicate that RIPK2 is required ...” (lines 147-148)

“Together, the findings suggest that RIPK2 positively ...” (lines 238-239)

“Collectively, the results suggest that RIPK2 stabilizes ...” (line 275)

“Together, the findings suggest that RIPK2 may ...” (line 332)

“Together, the results indicate that MKK7 is ...” (line 360)

“Together, the results suggest that both GSK583 ...” (line 408)

5. In the label free proteomics analyses, I can deduce that 6 replicates were used per sample (from the 24 raw files), but it would be good to explicitly state (at least in the Methods section) the number of replicates used, and if these were mere technical injections, or if they were from 6 individual cell cultures. Also, in addition to mentioning the use of FASP in the Methods section, more details are needed, such as approximate number of cells harvested per condition/replicate,

approximate amount of protein used for digestion, was trypsin the only enzyme used or some sort of LysC/trypsin combination, what filter units were used, etc.

We apologize for a lack of sufficient information. The 6 replicates per sample were derived from 3 biological replicates (*i.e.*, cells from 3 individual cell cultures), of which each was analyzed by LC-MS/MS twice (see Supplementary Fig. 8). The revised text (lines 626-634) now reads:

“Briefly, for each cell clone, three biological replicates of cultured cells ($\sim 5 \times 10^6$) were harvested, and cell pellets were lysed with about three volumes of lysis buffer (80 mM Tris-HCl, 4% SDS, 100 mM DTT, pH7.4). Cell lysates were sonicated to reduce the viscosity, and protein concentrations were measured using the Pierce 660 nm Protein Assay Kit. For each replicate, 50 μ g protein lysate was alkylated by 55 mM iodoacetamide and digested with trypsin (Promega, #V5280) in Microcon-YM30 spin filters (Millipore), using the filter-aid sample preparation (FASP) method. Each tryptic peptide sample was analyzed by liquid chromatography-tandem mass spectrometry (LC-MS/MS) twice.”

6. Results, page 7. “...we analyzed three PC3 single-cell clones”. When I first read single cell here, my mind wrongly jumped to single cell proteomics techniques. Of course, that is not what is used here. The single cell clones are surely expanded in culture, and then shotgun proteomics is carried out. But perhaps adding a few words here might help readers not to fall into this mental trap.

We modified the sentence to avoid confusion. Now the sentence (lines 174-176) reads:

“To identify consistent changes across different clones, three PC3 single-cell clones (#4, #12, and #16) with stable RIPK2-KO were isolated and expanded, and their proteomes were compared with those of control PC3 cells.”

7. Results, page 8. “Comprehensive proteomic analysis...” I don’t think “comprehensive” is correct here, given that only about 5000 proteins were identified/quantified. Sure, 5000 is a good analysis depth for single shot proteomics, but I think the word comprehensive should be reserved for prefractionated proteomics, where the number of proteins identified is closer to 10000.

We deleted “comprehensive” from the sentence (line 178).

8. In Supp. Fig. 10 one can see the fold change (FC) cut off used to determine up and downregulations in the label free proteomics. It was $+0.54$ (log₂ scale). How was that number determined? This FC cut off should be stated, if not in the main text, at least in the methods section. We apologize for the confusion. We did not set a fold change (FC) cut off for the 652 overlapping differentially expressed proteins (DEPs). In fact, 0.54 is the minimal value for the 652 DEPs in absolute value (see column CA of Table S2). To avoid confusion, we removed “0.54” and “-0.54” from Supplementary Fig. 10b.

9. Results, page 9. “...whereas RIPK2 overexpression significantly increased...” How was the RIPK2 overexpression carried out in cells? Was it by use of the RIPK2m4 plasmid, which one encounters downstream in the main text? Please clarify.

In the experiment, *RIPK2* (versus vector control) was stably transfected into parental PC3 cells. To avoid confusion, we revised the sentence. Now the text (line 204) reads:

“...whereas ectopic expression of human RIPK2 significantly increased ...”

Because parental PC3 cells (instead of *RIPK2*-KO PC3 cells) were used for transfection, we used unmodified *RIPK2* rather than *RIPK2m4*. In the figures, if unmodified *RIPK2* was used, we labeled the plasmid as “*RIPK2*”; if *RIPK2m4* was used, we labeled the plasmid as “*RIPK2m4*”.

10. Results, page 10. “...*RIPK2* and *MYC* signature genes...” Please define the gene signatures in the main text, not only in the legend of Fig.3g.

To define the gene signatures in the main text, we added new sentences (lines 209-212):

“To compute activity Z scores, we used the genes encoding the 243 protein groups downregulated by RIPK2-KO as RIPK2 signature genes. We also retrieved the Hallmark_MYC_Targets_V1 and V2 gene sets from the Molecular Signature Database (MSigDB)³⁵ and used them as MYC_V1 and MYC_V2 signature genes, respectively.”

11. Results, page 12. Please remove or substitute “Nevertheless”.

We removed “Nevertheless” (line 284).

12. Results, page 18. “...followed by daily treatment with vehicle control or GSK583.” Please state here the dosage applied (10 mg/kg/day, as stated in legend of Fig.8h).

We modified the text as the reviewer suggested. The text (line 428) now reads:

“...followed by daily treatment with vehicle control or GSK583 (10 mg/kg/day).”

13. In Methods, please correct PXD accession numbers. All are missing a 0 after PXD. Namely, PXD18890, PXD18870 and PXD18871 should be corrected to PXD018890, PXD018870 and PXD018871.

Thanking for spotting the errors! We have corrected all the PXD accession numbers in the revised manuscript.

REVIEWER COMMENTS

Reviewer #1 (Remarks to the Author):

The authors have addressed the concerns of the prior review.

Reviewer #2 (Remarks to the Author):

The authors have fully addressed my concern, and I recommend Accept at this stage.

Reviewer #4 (Remarks to the Author):

The authors have now appropriately addressed my earlier comments.

Reviewer #5 (Remarks to the Author):

In reference to the statistical and bioinformatics analysis I have the following comments:

1. I would suggest using a p-value for correlation cutoff along with the rho-cutoff (line 84, Fig 1a)
2. Are the p-values presented in Fig 1b nominal or adjusted? (this goes for all of the large-scale analysis that's done throughout)
3. I find the use of a barplots to show correlation values (3f and others) strange. I would try and find another way to visualize this since it is just one value and not showing any sort of distribution. Also, are there p-values associated with these correlations?
4. For Supp Fig 1b,c – how are these p-values calculated? Chi-square? Perhaps it would be more useful to show the amplification of these genes across the samples in the datasets in a heatmap or similar to show the percentage amplified in a way that is more intuitive. In general, I think the barplots in this paper that show one value per sample are a bit misleading.
5. Supp Fig 10. Are the volcano plots based on adjusted p values or nominal p?
6. Supp Fig 11 I suggest using an FDR not a p-value
7. Statistical analysis section in the methods is very brief. Were t-tests used for all of the tests?
8. I would remove the n.s. indications on the plots throughout – just indicate when something is significant

REVIEWER COMMENTS

Reviewer #1 (Remarks to the Author):

The authors have addressed the concerns of the prior review.

We thank the reviewer for agreeing that his/her previous concerns were addressed!

Reviewer #2 (Remarks to the Author):

The authors have fully addressed my concern, and I recommend Accept at this stage.

We thank the reviewer for recommending “Accept”!

Reviewer #4 (Remarks to the Author):

The authors have now appropriately addressed my earlier comments.

We thank the reviewer for agreeing that his/her earlier comments were appropriately addressed!

Reviewer #5 (Remarks to the Author):

In reference to the statistical and bioinformatics analysis I have the following comments:

1. I would suggest using a p-value for correlation cutoff along with the rho-cutoff (line 84, Fig 1a) Thanks for the suggestion! In the re-revised manuscript, we applied $\rho > 0.9$ and $p < 0.01$ as the cutoff values, resulting in the identification of 1,643 genes whose expression levels are associated with prostate cancer progression. Accordingly, we modified Figure 1a (also see Figure R1 below) as well as the source data for Figure 1a.

Figure R1. Venn diagram of human genes meeting the indicated criteria.

2. Are the p-values presented in Fig 1b nominal or adjusted? (this goes for all of the large-scale analysis that's done throughout)

Throughout the manuscript, *p*-values are nominal *p*-values, and *q*-values represent adjusted *p*-values computed using the Benjamini-Hochberg method.

3. I find the use of a barplots to show correlation values (3f and others) strange. I would try and find another way to visualize this since it is just one value and not showing any sort of distribution. Also, are there p-values associated with these correlations?

We now show a correlation matrix for RIPK2-induced activity scores, *RIPK2* mRNA levels, *MYC* mRNA levels, and *MYC* activity scores (V1 or V2) in the PCTA or TCGA (Firehose Legacy) PC cohorts (new Supplementary Fig. 14a, also see below for Figure R2). The matrix was generated in R-Studio version 4.1.0 using the `ggpairs` function in the `GGally` package (v2.1.2). Nominal *p*-values associated with the correlations are provided in the Source Data file (Fig. S14a).

Figure R2. Correlation matrix showing the correlations between *RIPK2* activity scores, *RIPK2* mRNA levels, *MYC* mRNA levels, *MYC_V1* activity scores, and *MYC_V2* activity scores in the PCTA (upper) and the TCGA (lower) PC cohorts. Numbers in upper triangle panels indicate Spearman correlation coefficients. Density plots show distributions of numeric variables (*i.e.*, *RIPK2* activity scores, *RIPK2* mRNA levels, *MYC* mRNA levels, *MYC_V1* activity scores, and *MYC_V2* activity scores). Scatter plots present the correlations between each pair of numeric variables. The correlation matrix was generated by R, using the `ggpairs` function in the `GGally` package (v2.1.2).

4. For Supp Fig 1b,c – how are these p-values calculated? Chi-square? Perhaps it would be more useful to show the amplification of these genes across the samples in the datasets in a heatmap or similar to show the percentage amplified in a way that is more intuitive. In general, I think the barplots in this paper that show one value per sample are a bit misleading.

Yes, the *p*-values were calculated using Chi-square. According to the reviewer’s suggestion, we replaced the Supplementary Figure 1b and 1c with a donut chart (new Supplementary Figure 1b; also see Figure R3 below) to show the distributions of the frequencies of *RIPK2* copy number alterations in different groups of PC samples (see Source Data for Fig. 1c).

Figure R3. Donut chart of the frequencies of copy number alterations of *RIPK2* in four different groups of PC samples. The numbers in the chart represent percentages. “None” indicates without *RIPK2* amplification or gain.

5. Supp Fig 10. Are the volcano plots based on adjusted p values or nominal p?

In Supplementary Figure 10a, the *p*-values are nominal *p*-values. Here, relatively loose criteria (nominal $p < 0.05$ and $\text{Log}_2\text{FC} > 0.5$ in absolute value) were applied to identify proteins differentially expressed in each *RIPK2*-KO PC3 stable cell clone (vs. control PC3 cells), prior to our overlapping analysis (Fig. 3a).

In Supplementary Figure 10b, the *p*-values in the y-axis are also nominal *p*-values. All the 409 consistently upregulated proteins and 243 consistently downregulated proteins have combined *p*-values of < 0.00187 , corresponding to *q*-values (*i.e.*, adjusted *p*-values) of < 0.005 (see Supplementary Table 2).

To avoid confusion, we modified the legend for Supplementary Figure 10 as follows.

- For panel a: “*P*-values represent nominal *p*-values.”
- For panel b: “The *y*-axis shows the negative \log_{10} -transformed combined nominal *p*-values.”

6. Supp Fig 11 I suggest using an FDR not a p-value

We revised the Supplementary Figure 11 as the reviewer suggested (also see Figure R4 below).

Figure R4. Bar graphs of the top 10 significantly enriched gene ontology (GO) terms identified by ToppFun analysis. a Bar plots of the top 10 significantly enriched biological processes (GO_BP). **b** Bar plots of the top 10 significantly enriched cellular components (GO_CC). For both panels, cyan and red represent significant enrichment by ToppFun analysis of the 243 downregulated and the 409 upregulated proteins, respectively.

7. Statistical analysis section in the methods is very brief. Were t-tests used for all of the tests? Per the guide for submission to *Nature Communications*, main text should be no more than 5,000 words. Therefore, we do not have space for a detailed description of all the statistical analyses. Nevertheless, all statistics and reproducibility information were reported in the figure legends.

8. I would remove the n.s. indications on the plots throughout – just indicate when something is significant.

We removed the n.s. indications on the plots throughout per the suggestion.

REVIEWER COMMENTS

Reviewer #5 (Remarks to the Author):

The authors have appropriately addressed my concerns.

REVIEWER COMMENTS

Reviewer #5 (Remarks to the Author):

The authors have appropriately addressed my concerns.

We thank the reviewer for agreeing that his/her concerns have been addressed!